# Self-Supervised Heterogeneous Graph Learning: a Homophily and Heterogeneity View

**Yujie Mo**[1,2]  **Feiping Nie**[3]  **Ping Hu**[1]  **Heng Tao Shen**[1]
**Zheng Zhang**[4*]  **Xinchao Wang**[2*]  **Xiaofeng Zhu**[1*]

[1]School of Computer Science and Engineering,
University of Electronic Science and Technology of China
[2]National University of Singapore
[3] School of Artificial Intelligence, Optics and Electronics (iOPEN),
Northwestern Polytechnical University
[4]School of Computer Science and Technology, Harbin Institute of Technology, Shenzhen

## Abstract

Self-supervised heterogeneous graph learning has achieved promising results in various real applications, but still suffers from the following issues: (i) meta-paths can be employed to capture the homophily in the heterogeneous graph, but meta-paths are human-defined, requiring substantial expert knowledge and computational costs; and (ii) the heterogeneity in the heterogeneous graph is usually underutilized, leading to the loss of task-related information. To solve these issues, this paper proposes to capture both homophily and heterogeneity in the heterogeneous graph without pre-defined meta-paths. Specifically, we propose to learn a self-expressive matrix to capture the homophily from the subspace and nearby neighbors. Meanwhile, we propose to capture the heterogeneity by aggregating the information of nodes from different types. We further design a consistency loss and a specificity loss, respectively, to extract the consistent information between homophily and heterogeneity and to preserve their specific task-related information. We theoretically analyze that the learned homophilous representations exhibit the grouping effect to capture the homophily, and considering both homophily and heterogeneity introduces more task-related information. Extensive experimental results verify the superiority of the proposed method on different downstream tasks.

## 1 Introduction

Heterogeneous graph learning aims to extract and uncover meaningful hidden patterns in the heterogeneous graph, such that it outputs discriminative representations for different tasks (Dong et al., 2017; Sun et al., 2021). To alleviate the issue of limited labels in real scenarios, self-supervised heterogeneous graph learning (SHGL) has received increasing attention across diverse applications, such as social network analysis and recommendation systems (Chen et al., 2022; Xie et al., 2022).

Existing SHGL methods typically employ meta-paths to extract semantic relationships among nodes of the same type in the heterogeneous graph, as illustrated in Figure 1(a). Consequently, such a process treats the heterogeneous graph as a composition of homogeneous graphs based on meta-paths (Wang et al., 2023; Mo et al., 2023a). This actually mines the **homophily** (*i.e.,* connectivity and information aggregation among nodes within the same class) in the heterogeneous graph, as two nodes connected by meta-path generally tend to belong to the same class. For instance, in an academic heterogeneous graph with several node types (*e.g.,* author, paper, and subject), if there exists a meta-path "paper-author-paper" between two papers (*i.e.,* two papers are written by the same author), these two papers possibly belong to the same class. As a result, previous SHGL methods utilize meta-paths to explore the homophily, increasing the intra-class correlation and benefiting downstream tasks, as shown in Figure 1(b).

---

*Corresponding authors: X. Zhu, X. Wang, and Z. Zhang

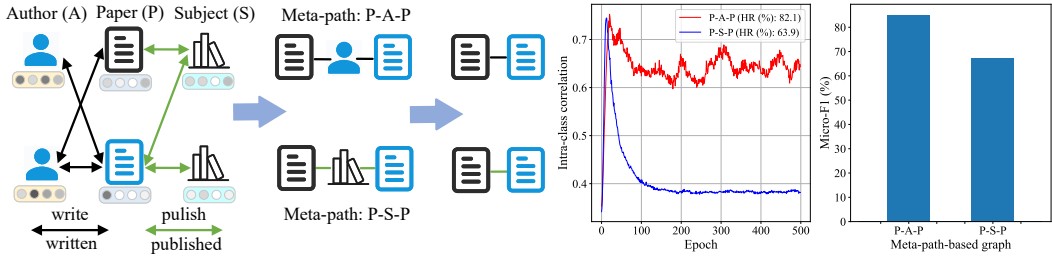

(a) Example of meta-path-based graphs      (b) Studies of different meta-path-based graphs

Figure 1: Example and studies of meta-path-based graphs in previous SHGL. (a) For an academic heterogeneous graph, most previous SHGL employs meta-paths (*e.g.,* P-A-P and P-S-P) to establish connections between two papers, and then ignores nodes from other types (*e.g.,* Author) in meta-paths. (b) Intra-class correlation and node classification results (*i.e.,* Micro-F1) by GCN (Kipf & Welling, 2017) on meta-path-based homogeneous graphs with different homophily ratios (HR, *i.e.,* the ratio of nodes connected by meta-path belong to the same class). The higher the HR of the meta-path-based graph, the higher the intra-class correlation, thus benefiting the classification performance.

However, existing SHGL methods still have limitations that need to be addressed. On the one hand, meta-paths are manually defined and require expert knowledge to select appropriate meta-paths for different tasks (Lv et al., 2021). Moreover, employing meta-paths to extract the relationships among nodes incurs considerable computation costs, which exponentially increase with the meta-path length. On the other hand, most previous SHGL methods overlook or cannot effectively utilize the **heterogeneity** (*i.e.,* connectivity and information aggregation among nodes from different types) in the heterogeneous graph, which may carry significant information relevant to downstream tasks. Take the same academic heterogeneous graph as an example. If two authors have the same name in the same institution, it is difficult to distinguish these two authors. In contrast, if we consider the heterogeneity (*e.g.,* each author's published papers), we can easily distinguish them. As a result, previous SHGL methods may lose significant task-related information associated with the heterogeneity.

Based on the above analysis, it is feasible to consider both homophily and heterogeneity in the heterogeneous graph without pre-defined meta-paths to improve the effectiveness of SHGL. To achieve this, there are at least two challenges to be solved, *i.e.,* (i) capturing the homophily in the heterogeneous graph without relying on meta-paths; and (ii) effectively utilizing both homophily and heterogeneity in the heterogeneous graph, despite their inherent conflict.

In this paper, to address the above challenges, we discard traditional meta-paths and propose a novel SHGL framework to capture both Homophily and hetEROgeneity in the heterogeneous graph (HERO for short), as shown in Figure 2. Specifically, we obtain the closed-form solution of the self-expressive matrix to capture the homophily from the subspace and nearby neighbors and obtain homophilous representations, thus tackling **Challenge (i)**. Meanwhile, we employ a heterogeneous encoder to aggregate the information of nodes from different types to capture the heterogeneity and thereby obtain heterogeneous representations. With homophilous and heterogeneous representations, we further design a consistency loss and a specificity loss to capture the invariance between them and preserve their respective task-related information in the latent space, respectively, thus tackling **Challenge (ii)**. Finally, in theoretical analysis, the learned homophilous representations are proved to capture the homophily in the heterogeneous graph, while homophilous and heterogeneous representations are fused to introduce more information related to downstream tasks.

Compared to previous SHGL methods, our main contributions can be summarized as follows:

- We make the first attempt to understand the self-supervised heterogeneous graph learning without pre-defined meta-paths from the view of homophily and heterogeneity.

- We propose to comprehensively capture the homophily from both the subspace and nearby neighbors as well as to discard pre-defined meta-paths that require expert knowledge. We further extract consistent and specific information between homophilous and heterogeneous representations to introduce more task-related information, thus achieving effectiveness.

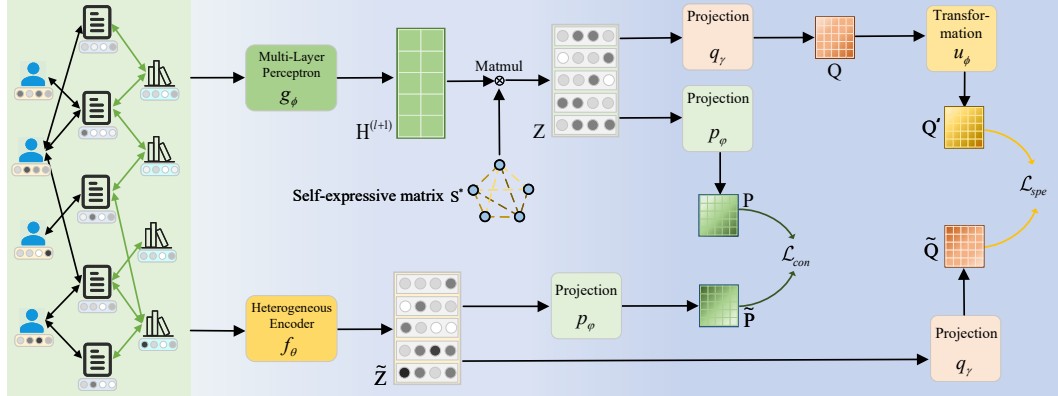

Figure 2: The flowchart of the proposed HERO. Specifically, HERO first employs the Multi-Layer Perception as encoder $g_\phi$ and learns a self-expressive matrix $\mathbf{S}^*$ to capture the homophily and obtain homophilous representations $\mathbf{Z}$. Meanwhile, HERO employs a heterogeneous encoder $f_\theta$ to aggregate the information of nodes from different types to obtain heterogeneous representations $\widetilde{\mathbf{Z}}$. After that, HERO designs a consistency loss $\mathcal{L}_{con}$ and a specificity loss $\mathcal{L}_{spe}$ to extract the consensus between $\mathbf{Z}$ and $\widetilde{\mathbf{Z}}$ as well as to maintain their distinct information in different latent spaces, respectively.

- We theoretically demonstrate that the learned homophilous representations have the grouping effect, thus capturing the homophily. Furthermore, we theoretically demonstrate that considering both homophily and heterogeneity introduces more task-related information than considering them individually, thus benefiting downstream tasks.

- We experimentally demonstrate the superiority of the proposed method in terms of different downstream tasks on both heterogeneous graph datasets and homogeneous graph datasets, compared to numerous comparison methods.

## 2 METHOD

**Notations.** Let $\mathbf{G} = (\mathcal{V}, \mathcal{E}, \mathbf{X}, \mathcal{T}, \mathcal{R})$ represent a heterogeneous graph, where $\mathcal{V} = \{v_i\}_{i=1}^N$ and $\mathcal{E}$ indicate nodes set and edges set, respectively, and $N$ indicates the number of nodes. $\mathbf{X} = \{\mathbf{x}_i\}_{i=1}^N$ denotes the node features matrix, while $\mathcal{T}$ and $\mathcal{R}$ indicate node types set and edge types set, respectively. Given the heterogeneous graph, the meta-path used in previous SHGL methods can be defined in the form of $v_1 \xrightarrow{r_1} v_2 \xrightarrow{r_2} \cdots \xrightarrow{r_s} v_{s+1}$. It is a sequence of a composite relation $r_1 \circ r_2 \circ \cdots \circ r_s$ between node $v_1$ and node $v_{s+1}$, where $s$ indicates the length of meta-path and $\circ$ denotes the composition operator. Many previous SHGL methods explore the homophily in the heterogeneous graph with different meta-paths, as shown in Figure 1(a). In contrast, the proposed method mines both homophily and heterogeneity in the heterogeneous graph without pre-defined meta-paths, as shown in Figure 2, and we introduce the details of the proposed method as follows.

### 2.1 HOMOPHILY

In this paper, we propose to adaptively learn the homophily in the heterogeneous graph without meta-paths. Actually, the homophily extraction in the heterogeneous graph aims at establishing connections and conducting information aggregation among nodes within the same class. Considering the learning scenario without labels, previous studies (Chapelle et al., 2002; Zhou et al., 2003) show two statements: (i) nodes in the same subspace are likely to belong to the same class; and (ii) nearby nodes are likely to belong to the same class. Hence, there are at least two ways for the homophily extraction in the heterogeneous graph. The first way is to connect and aggregate the information of nodes in the same subspace, while the second way is to connect and aggregate the information of each node and its nearby neighbors.

In this paper, we first employ the Multi-Layer Perceptron (MLP) as the encoder $g_\phi$ to obtain the $(l+1)$-th layer node representations $\mathbf{H}^{(l+1)}$ by:

$$\mathbf{H}^{(l+1)} = \sigma(\mathbf{H}^{(l)}\mathbf{W}^{(l)}), \tag{1}$$

where $\sigma$ is the activation function, $\mathbf{W}^{(l)}$ indicates the trainable parameters of $g_\phi$, and $\mathbf{H}^{(0)} = \mathbf{X}$. After that, we propose to capture the homophily in the subspace with a self-expressive matrix $\mathbf{S} \in \mathbb{R}^{N \times N}$ that linearly describes every node by all nodes, *i.e.,*

$$\mathbf{H}^{(l+1)} = \mathbf{S}\mathbf{H}^{(l+1)} + \mathbf{O}^{(l+1)}, \tag{2}$$

where $\mathbf{O}^{(l+1)}$ is a noise matrix. In Eq. (2), the representation of the $i$-th node $\mathbf{h}_i^{(l+1)}$ can be represented by $\mathbf{h}_i^{(l+1)} = s_{i1}\mathbf{h}_1^{(l+1)} + \cdots + s_{iN}\mathbf{h}_N^{(l+1)}$. In particular, the larger the weight (*i.e.,* the value of $s_{ij}$), the higher the probability of the node $v_i$ replaced by the node $v_j$. According to the subspace-preserving property (He et al., 2003; Vidal, 2009), each node and its corresponding nodes with large weights are likely to fall in the same subspace, so they are likely to belong to the same class. Therefore, the self-expressive matrix describes each node by the nodes (*i.e.,* with large weights) in the same subspace to capture the homophily.

However, the self-expressive matrix may also introduce the nodes from different classes (*i.e.,* with small weights) into the subspace to degrade the model performance. A good solution is to push the weight of nodes from different classes to be as small as possible. Based on the second statement, we propose to encourage the self-expressive matrix to focus more on the neighbors of each node in the original feature space and less on its faraway nodes that may come from different classes. To do this, we first calculate the feature distance matrix $\mathbf{D} \in \mathbb{R}^{N \times N}$ between all node pairs, where $d_{ij} = ||\mathbf{x}_i - \mathbf{x}_j||_2^2$, and then treat the value less than the threshold as 0 to obtain a sparse feature distance matrix. Based on the above analysis, each node $v_i$ and its neighbors set $\mathcal{N}_i$ ($\forall v_j \in \mathcal{N}_i$, $d_{ij} = 0$) should be further assigned with large weights in the self-expressive matrix while the weights of faraway nodes should be penalized. To achieve this, we propose to capture the homophily from both the subspace and nearby neighbors by:

$$\min_{\mathbf{S}} \|\mathbf{H}^{(l+1)} - \mathbf{S}\mathbf{H}^{(l+1)}\|_F^2 + \alpha \sum_{i,j=1}^N d_{ij}s_{ij} + \beta \sum_{i,j=1}^N s_{ij}^2, \tag{3}$$

where $\alpha$ and $\beta$ are non-negative parameters to trade off three terms. In Eq. (3), the second term enables the self-expressive matrix to focus on the nearby neighbors (*i.e.,* with small feature distance) of each node to capture the homophily. Moreover, the second term penalizes the weights among nodes from different classes induced by the first term, and the first term captures part connections within the same class missed by the second term, thus complementing each other. The third term aims to regularize the self-expressive matrix $\mathbf{S}$ to avoid the trivial solution. Actually, Eq. (3) takes the closed-form solution $\mathbf{S}^*$ as follows:

$$\mathbf{S}^* = (\mathbf{H}^{(l+1)}(\mathbf{H}^{(l+1)})^T - \frac{\alpha}{2}\mathbf{D})(\mathbf{H}^{(l+1)}(\mathbf{H}^{(l+1)})^T + \beta\mathbf{I}_N)^{-1}, \tag{4}$$

where $\mathbf{I}_N$ is the identity matrix. In fact, the self-expressive matrix shares a similar idea with the widely used self-attention mechanism (Vaswani et al., 2017), *i.e.,* linearly describe each sample with all samples. However, there are major differences between them, and we list them in Appendix A.3.

After achieving the closed-form solution $\mathbf{S}^*$, we conduct information aggregation among nodes of the same type in the heterogeneous graph to obtain homophilous representations $\mathbf{Z}$, *i.e.,*

$$\mathbf{Z} = \mathbf{S}^*\mathbf{H}^{(l+1)}. \tag{5}$$

However, directly obtaining $\mathbf{Z}$ by Eq. (5) incurs expensive computation costs due to the cubic time complexity in computing $\mathbf{S}^*$ in Eq. (4) and the quadratic time complexity in calculating $\mathbf{S}^*\mathbf{H}^{(l+1)}$. To alleviate this issue, we avoid directly calculating $\mathbf{S}^*$ and reformulate Eq. (5) by the matrix identity transformation (Woodbury, 1950) (details are shown in Appendix C.3), *i.e.,*

$$\mathbf{Z} = \mathbf{H}^{(l+1)}(\mathbf{H}^{(l+1)})^T\mathbf{B} - \frac{\alpha}{2}\mathbf{D}\mathbf{B}, \tag{6}$$

where $\mathbf{B} = \frac{1}{\beta}\mathbf{H}^{(l+1)} - \frac{1}{\beta^2}\mathbf{H}^{(l+1)}(\mathbf{I}_d + \frac{1}{\beta}(\mathbf{H}^{(l+1)})^T\mathbf{H}^{(l+1)})^{-1}(\mathbf{H}^{(l+1)})^T\mathbf{H}^{(l+1)}$, and $\mathbf{I}_d \in \mathbb{R}^{d \times d}$ is the identity matrix, where $d$ indicates the dimension of node representations. By reordering the

matrix multiplication, we further reduce the time complexity of Eq. (6) to $\mathcal{O}(Nd^2 + d^3 + kd)$, where $d^2 \ll N$ and $k \ll N^2$, and $k$ indicates the nonzero entries of the sparse feature distance matrix $\mathbf{D}$, details are shown in Appendix B.2.

Therefore, the proposed method is available to capture the homophily in the heterogeneous graph in an effective and efficient way. In this paper, we further prove that both the self-expressive matrix $\mathbf{S}^*$ and the learned homophilous representations $\mathbf{Z}$ capture the homophily by having the grouping effect. To do this, we first follow (Li et al., 2020) to define the grouping effect as follows.

**Definition 2.1.** *Given the nodes set $\mathcal{V} = \{v_i\}_{i=1}^N$, if $|c_{ik} - c_{jk}| \to 0$ ($\forall 1 \le k \le F'$) holds for every $v_i$ and $v_j$ satisfying $v_i \to v_j$ (i.e., $\|\mathbf{x}_i - \mathbf{x}_j\|_2 \to 0$), the matrix $\mathbf{C} \in \mathbb{R}^{N \times F'}$ has the grouping effect.*

Based on Definition 2.1, if a matrix $\mathbf{C}$ has the grouping effect and the condition $v_i \to v_j$ holds for two nodes $v_i$ and $v_j$, then every element of the $i$-th and the $j$-th row ($\mathbf{c}_i$ and $\mathbf{c}_j$) should be aligned. Indeed, the condition $v_i \to v_j$ indicates two nodes may belong to the same class, thus the alignment between $\mathbf{c}_i$ and $\mathbf{c}_j$ reflects the homophily among nodes. After that, we follow Definition 2.1 to prove the grouping effect of the self-expressive matrix $\mathbf{S}^*$ and homophilous representations $\mathbf{Z}$ by Theorem 2.2, whose proof can be found in Appendix C.4.

**Theorem 2.2.** *Both self-expressive matrix $\mathbf{S}^* \in \mathbb{R}^{N \times N}$ and homophilous representations $\mathbf{Z} \in \mathbb{R}^{N \times d}$ have the grouping effect for any two nodes $v_i$ and $v_j$ that hold the condition $v_i \to v_j$, i.e.,*

$$v_i \to v_j \Rightarrow \left| s_{ip}^* - s_{jp}^* \right| \to 0, \text{ and } |z_{iq} - z_{jq}| \to 0, \forall 1 \le q \le d, 1 \le i,j,p \le N. \tag{7}$$

Based on Theorem 2.2, if two nodes $v_i$ and $v_j$ have similar node features, *i.e.,* being likely to belong to the same class, both their self-expressive vectors (*i.e.,* $\mathbf{s}_i^*$ and $\mathbf{s}_j^*$) and representations (*i.e.,* $\mathbf{z}_i$ and $\mathbf{z}_j$) are expected to be similar. As a result, both $\mathbf{S}^*$ and $\mathbf{Z}$ have the grouping effect, thus capturing the homophily in the heterogeneous graph (verified in Section 3.2.3).

## 2.2 HETEROGENEITY

In addition to the homophily, the heterogeneity is also significant for the heterogeneous graph as it may contain task-related information Zhang et al. (2022a). However, most existing SHGL methods overlook or cannot effectively utilize the heterogeneity from nodes of different types. As a result, previous SHGL methods may lose discriminative information in the heterogeneity to induce a negative impact on downstream tasks. Therefore, it is necessary to capture the heterogeneity in the heterogeneous graph to improve the effectiveness of SHGL.

To do this, we propose to aggregate the information of nodes from different types in the heterogeneous graph. Specifically, for the node $v_i$, we employ a heterogeneous encoder $f_\theta$ to aggregate the information of its relevant one-hop neighbors (*i.e.,* nodes of other types) based on the edge type $r \in \mathcal{R}$, and then obtain the edge-based representations $\tilde{\mathbf{z}}_{i,r}^{(l+1)}$ by:

$$\tilde{\mathbf{z}}_{i,r}^{(l+1)} = \delta\left(\left(\frac{1}{m}\sum_{j=1}^m \left\{\tilde{\mathbf{z}}_j^{(l)} \mid v_j \in \mathcal{N}_{i,r}\right\}\right)\mathbf{W}_r^{(l)}\right), \tag{8}$$

where $\delta$ indicates the activation function, $\mathcal{N}_{i,r}$ indicates the one-hop neighbors set of the node $v_i$ based on the edge type $r$, $m$ is the number of the neighbors, and $\mathbf{W}_r^{(l)}$ indicates the trainable parameters of $f_\theta$. Considering all edge types in the heterogeneous graph, we further obtain the heterogeneous representations by fusing all edge-based representations, *i.e.,*

$$\widetilde{\mathbf{Z}}^{(l+1)} = \frac{1}{|\mathcal{R}|}\sum_{r \in \mathcal{R}} \widetilde{\mathbf{Z}}_r^{(l+1)}, \tag{9}$$

where $|\mathcal{R}|$ indicates the number of edge types. Finally, we use $\widetilde{\mathbf{Z}}$ to represent the last layer of heterogeneous representations $\widetilde{\mathbf{Z}}^{(L)}$ for brief, where $L$ is the number of layers. As a result, the heterogeneous representations $\widetilde{\mathbf{Z}}$ aggregate the information of nodes from different types and thus are expected to capture the heterogeneity in the heterogeneous graph.

## 2.3 CONNECTIONS BETWEEN HOMOPHILY AND HETEROGENEITY

Given homophilous representations $\mathbf{Z}$ and heterogeneous representations $\widetilde{\mathbf{Z}}$, we have the following observations: (i) they are both representations of the same node, sharing the same original node features. Therefore, they are intuitive to contain the consistent information; and (ii) the homophilous representations focus on aggregating the information from nodes of the same class, while the heterogeneous representations focus on aggregating the information from nodes of different types. As a result, homophilous and heterogeneous representations contain specific information within each of them, respectively. To effectively utilize homophilous and heterogeneous representations (*i.e.,* $\mathbf{Z}$ and $\widetilde{\mathbf{Z}}$), we design a consistency loss and a specificity loss to extract the consistent information between them and to maintain their individually specific information related to downstream tasks, respectively.

Specifically, we first propose to learn a projection head $p_\varphi$ to map both homophilous representations and heterogeneous representations into the same latent space, *i.e.,* $\mathbf{P} = p_\varphi(\mathbf{Z})$ and $\widetilde{\mathbf{P}} = p_\varphi(\widetilde{\mathbf{Z}})$. After that, we design a consistency loss to maximize the invariance between $\mathbf{P}$ and $\widetilde{\mathbf{P}}$ by:

$$\mathcal{L}_{\text{con}} = \sum_{n=1}^{N} (\mathbf{p}_n - \tilde{\mathbf{p}}_n)^2 + \gamma \log\big( \sum_{i,j=1}^{d} e^{\sum_{n=1}^{N} (p_{ni}p_{nj} + \tilde{p}_{ni}\tilde{p}_{nj})} \big), \tag{10}$$

where $i$ and $j$ indicate $i$-th and $j$-th dimensions of $\mathbf{p}_n$, respectively, and $\gamma$ is a non-negative parameter. In Eq. (10), the first term encourages both $\mathbf{P}$ and $\widetilde{\mathbf{P}}$ to agree with each other, thus converging to the consistency. The second term enforces different dimensions of $\mathbf{P}$ and $\widetilde{\mathbf{P}}$ to uniformly distribute over the latent space, thus avoiding the issue of model collapse. As a result, Eq. (10) is available to extract the consistent information between homophilous representations and heterogeneous representations.

In addition to extracting the consistent information, we aim to preserve the distinct characteristics of homophilous and heterogeneous representations as well as maintain their task-related information. However, we cannot directly add such regularization or constraints on both $\mathbf{P}$ and $\widetilde{\mathbf{P}}$, as its goal differs from that of Eq. (10) and may lead to conflicts. To solve this issue, we propose to learn a projection head $q_\gamma$ to map both homophilous representations and heterogeneous representations into another latent space, *i.e.,* $\mathbf{Q} = q_\gamma(\mathbf{Z})$ and $\widetilde{\mathbf{Q}} = q_\gamma(\widetilde{\mathbf{Z}})$. After that, we employ a transformation head $u_\phi$ on $\mathbf{Q}$ to obtain $\mathbf{Q}' = u_\phi(\mathbf{Q})$. We further design a specificity loss to preserve the specific information related to homophilous representations and heterogeneous representations by:

$$\mathcal{L}_{\text{spe}} = \sum_{n=1}^{N} (\mathbf{q}'_n - \tilde{\mathbf{q}}_n)^2 - \eta \sum_{n=1}^{N} ((\mathbf{q}'_n - \mathbf{q}_n)^2 + (\mathbf{q}'_n - \mathbf{p}_n)^2), \tag{11}$$

where $\eta$ is a non-negative parameter. In Eq. (11), the first term aims to align $\widetilde{\mathbf{Q}}$ with $\mathbf{Q}'$, instead of aligning $\widetilde{\mathbf{Q}}$ with $\mathbf{Q}$. As a result, it avoids directly aligning the projection of homophilous representations and heterogeneous representations to preserve their distinct information. Actually, if $u_\phi$ is an identity transformation, the first term in Eq. (11) is the same as the first term in Eq. (10). In addition, even if $u_\phi$ is not an identity transformation, $\mathbf{Q}'$ may also be equal to $\mathbf{P}$, leading to the redundancy with the consistency loss. To avoid such scenarios, the second term in Eq. (11) enforces the transformed $\mathbf{Q}'$ different from the original $\mathbf{Q}$ and $\mathbf{P}$. As a result, Eq. (11) maintains the respective task-related information of homophilous representations and heterogeneous representations.

We integrate the consistency loss with the specificity loss to have the final objective function as:

$$\mathcal{J} = \mathcal{L}_{con} + \lambda \mathcal{L}_{spe}, \tag{12}$$

where $\lambda$ is a non-negative parameter. Finally, we concatenate homophilous representations $\mathbf{Z}$ with heterogeneous representations $\widetilde{\mathbf{Z}}$ to obtain final representations $\hat{\mathbf{Z}}$ for downstream tasks. As a result, the concatenated representations $\hat{\mathbf{Z}}$ considering both homophily and heterogeneity in the heterogeneous graph can be theoretically proved to introduce more task-related information by Theorem 2.3, whose proof can be found in Appendix C.5.

**Theorem 2.3.** *For any downstream task $T$, the representations with both homophily and heterogeneity (e.g., $\hat{\mathbf{Z}}$) contain more task-related information than the representations with only homophily (e.g., $\mathbf{Z}$) or with only heterogeneity (e.g., $\widetilde{\mathbf{Z}}$), i.e.,*

$$I(\hat{\mathbf{Z}}, T) \geq \max(I(\mathbf{Z}, T), I(\widetilde{\mathbf{Z}}, T)), \tag{13}$$

*where $I(\cdot, \cdot)$ indicates the mutual information.*

Table 1: Classification performance (*i.e.,* Macro-F1 and Micro-F1) on heterogeneous graph datasets.

| Method | ACM | | Yelp | | DBLP | | Aminer | |
|---|---|---|---|---|---|---|---|---|
| | Macro-F1 | Micro-F1 | Macro-F1 | Micro-F1 | Macro-F1 | Micro-F1 | Macro-F1 | Micro-F1 |
| Deep Walk | 73.9±0.3 | 74.1±0.1 | 68.7±1.1 | 73.2±0.9 | 88.1±0.2 | 89.5±0.3 | 54.7±0.8 | 59.7±0.7 |
| GCN | 86.9±0.2 | 87.0±0.3 | 85.0±0.6 | 87.4±0.8 | 90.2±0.2 | 90.9±0.5 | 64.5±0.7 | 71.5±0.9 |
| GAT | 85.0±0.4 | 84.9±0.3 | 86.4±0.5 | 88.2±0.7 | 91.0±0.4 | 92.1±0.2 | 63.8±0.4 | 70.6±0.7 |
| Mp2vec | 87.6±0.5 | 88.1±0.3 | 78.2±0.8 | 83.6±0.9 | 85.7±0.3 | 87.6±0.6 | 58.7±0.5 | 65.3±0.6 |
| HAN | 89.4±0.2 | 89.2±0.2 | 90.5±1.2 | 90.7±1.4 | 91.2±0.4 | 92.0±0.5 | 65.3±0.7 | 72.8±0.4 |
| HGT | 91.5±0.7 | 91.6±0.6 | 89.9±0.5 | 90.2±0.6 | 90.9±0.6 | 91.7±0.8 | 64.5±0.5 | 71.0±0.7 |
| DMGI | 89.8±0.1 | 89.8±0.1 | 82.9±0.8 | 85.8±0.9 | 92.1±0.2 | 92.9±0.3 | 63.8±0.4 | 67.6±0.5 |
| DMGIattn | 88.7±0.3 | 88.7±0.5 | 82.8±0.7 | 85.4±0.5 | 90.9±0.2 | 91.8±0.3 | 62.4±0.9 | 66.8±0.8 |
| HDMI | 90.1±0.3 | 90.1±0.3 | 80.7±0.6 | 84.0±0.9 | 91.3±0.2 | 92.2±0.5 | 65.9±0.4 | 71.7±0.6 |
| HeCo | 88.3±0.3 | 88.2±0.2 | 85.3±0.7 | 87.9±0.6 | 91.0±0.3 | 91.6±0.2 | 71.8±0.9 | 78.6±0.7 |
| HGCML | 90.6±0.7 | 90.7±0.5 | 90.7±0.8 | 91.0±0.7 | 91.9±0.8 | 93.2±0.7 | 70.5±0.4 | 76.3±0.6 |
| CPIM | 91.4±0.3 | 91.3±0.2 | 90.2±0.5 | 90.3±0.4 | 93.2±0.6 | 93.8±0.8 | 70.1±0.9 | 75.8±1.1 |
| HGMAE | 90.5±0.5 | 90.6±0.7 | 90.5±0.7 | 90.7±0.5 | 92.9±0.5 | 93.4±0.6 | 72.3±0.9 | 80.3±1.2 |
| DMG | 91.0±0.3 | 90.9±0.4 | 90.8±0.5 | 91.2±0.6 | 93.3±0.2 | 94.0±0.3 | 72.0±0.7 | 79.5±0.9 |
| **HERO** | **92.2±0.5** | **92.1±0.7** | **92.4±0.7** | **92.3±0.6** | **93.8±0.6** | **94.4±0.4** | **75.1±0.7** | **84.5±0.9** |

Theorem 2.3 indicates that considering both homophily and heterogeneity introduces more task-related information than considering them individually, to benefit downstream tasks (verified by the Corollary in Appendix C.6). Therefore, the proposed method is expected to perform better on different downstream tasks than previous SHGL methods that consider only the homophily in the heterogeneous graph (verified in Section 3.2).

## 3 EXPERIMENTS

In this section, we conduct experiments on both heterogeneous and homogeneous graph datasets to evaluate the proposed HERO in terms of different downstream tasks (*i.e.,* node classification and similarity search), compared to heterogeneous and homogeneous graph methods. Detailed settings are shown in Appendix D. Additional experimental results are shown in Appendix E.

### 3.1 EXPERIMENTAL SETUP

#### 3.1.1 DATASETS

The used datasets include five heterogeneous graph datasets and four homogeneous graph datasets. Heterogeneous graph datasets include three academic datasets (*i.e.,* ACM (Wang et al., 2019), DBLP (Wang et al., 2019), and Aminer (Hu et al., 2019)), one business dataset (*i.e.,* Yelp (Lu et al., 2019)), and one huge knowledge graph dataset (*i.e.,* Freebase (Lv et al., 2021)). Homogeneous graph datasets include two sale datasets (*i.e.,* Amazon-Photo and Amazon-Computers (Shchur et al., 2018)), and two co-authorship datasets (*i.e.,* Coauther-CS and Coauther-Physics (Sinha et al., 2015)).

#### 3.1.2 COMPARISON METHODS

The comparison methods include eleven heterogeneous graph methods and twelve homogeneous graph methods. The former includes two semi-supervised methods (*i.e.,* HAN (Wang et al., 2019) and HGT (Hu et al., 2020)), one traditional unsupervised method (*i.e.,* Mp2vec (Dong et al., 2017)), and eight self-supervised methods (*i.e.,* DMGI (Park et al., 2020), DMGIattn (Park et al., 2020), HDMI (Jing et al., 2021), HeCo (Wang et al., 2021), HGCML (Wang et al., 2023), CPIM (Mo et al., 2023b), HGMAE (Tian et al., 2023), and DMG (Mo et al., 2023a)). The latter includes two semi-supervised methods (GCN (Kipf & Welling, 2017) and GAT (Velickovic et al., 2018)), one traditional unsupervised method (*i.e.,* DeepWalk (Perozzi et al., 2014)), and nine self-supervised methods, (*i.e.,* DGI (Velickovic et al., 2019), GMI (Peng et al., 2020), MVGRL (Hassani & Khasahmadi, 2020), GRACE (Zhu et al., 2020b), GCA (Zhu et al., 2021), GIC (Mavromatis & Karypis, 2021), G-BT (Bielak et al., 2022), COSTA (Zhang et al., 2022b), and DSSL (Xiao et al., 2022)).

Table 2: Classification performance (*i.e.,* Macro-F1 and Micro-F1) on homogeneous graph datasets, where OOM indicates Out-Of-Memory.

| Method | Amazon-Photo | | Amazon-Computers | | Coauther-CS | | Coauther-Physics | |
|---|---|---|---|---|---|---|---|---|
| | Macro-F1 | Micro-F1 | Macro-F1 | Micro-F1 | Macro-F1 | Micro-F1 | Macro-F1 | Micro-F1 |
| Deep Walk | 87.4±0.5 | 89.7±0.3 | 84.0±0.3 | 85.6±0.4 | 81.1±0.5 | 84.6±0.7 | 90.4±0.6 | 91.8±0.5 |
| GCN | 90.5±0.3 | 92.5±0.2 | 84.0±0.4 | 86.4±0.3 | 90.1±0.8 | 93.0±0.5 | 93.8±0.6 | 95.6±0.6 |
| GAT | 90.2±0.5 | 91.8±0.4 | 83.2±0.2 | 85.7±0.4 | 89.7±0.3 | 92.3±0.5 | 93.6±0.7 | 95.5±0.5 |
| DGI | 89.3±0.2 | 91.6±0.3 | 79.3±0.3 | 83.9±0.5 | 89.4±0.6 | 92.2±0.8 | 93.2±0.7 | 94.5±0.5 |
| GMI | 89.3±0.4 | 90.6±0.2 | 80.1±0.4 | 82.2±0.4 | OOM | OOM | OOM | OOM |
| MVGRL | 90.1±0.3 | 91.7±0.4 | 84.6±0.6 | 86.9±0.5 | 89.3±0.4 | 92.1±0.7 | 93.6±0.6 | 95.3±0.7 |
| GRACE | 90.3±0.5 | 91.9±0.3 | 84.2±0.3 | 86.8±0.5 | 90.2±0.8 | 93.0±0.6 | 94.2±0.5 | 95.7±0.4 |
| GCA | 91.1±0.4 | 92.4±0.4 | 85.9±0.5 | 87.7±0.3 | 90.1±0.3 | 92.9±0.5 | 94.1±0.6 | 95.7±0.3 |
| GIC | 90.0±0.3 | 91.6±0.2 | 82.6±0.4 | 84.9±0.3 | 87.3±0.4 | 90.5±0.6 | 93.1±0.3 | 93.9±0.4 |
| G-BT | 91.5±0.4 | 92.6±0.6 | 86.2±0.3 | 88.1±0.5 | 90.0±0.7 | 93.0±0.5 | 93.0±0.6 | 95.1±0.4 |
| COSTA | 91.3±0.4 | 92.5±0.3 | **86.4±0.3** | 88.3±0.4 | 90.1±0.6 | 93.0±0.7 | 94.0±0.4 | 95.7±0.6 |
| DSSL | 90.6±0.2 | 92.1±0.3 | 85.6±0.3 | 87.3±0.4 | 89.5±0.3 | 92.2±0.4 | 93.3±0.5 | 95.2±0.4 |
| **HERO** | **91.8±0.4** | **93.0±0.3** | 85.7±0.6 | **88.4±0.5** | **90.6±0.5** | **93.3±0.6** | **94.6±0.7** | **96.0±0.5** |

For a fair comparison, we follow (Dong et al., 2017; Wang et al., 2019; Lu et al., 2019; Lv et al., 2021) to select meta-path-based graphs for previous meta-path-based SHGL methods. Moreover, we follow (Mo et al., 2023a) to conduct homogeneous graph methods on heterogeneous graph datasets by separately learning the representations of each meta-path-based graph and further concatenating them for downstream tasks. In addition, we replace the heterogeneous encoder $f_\theta$ with GCN to implement the proposed method on homogeneous graph datasets because there is only one node type in the homogeneous graph. The code is released at `https://github.com/YujieMo/HERO`.

## 3.2 RESULTS ANALYSIS

### 3.2.1 EFFECTIVENESS ON THE HETEROGENEOUS GRAPH

We first evaluate the effectiveness of the proposed method on the heterogeneous graph datasets by reporting the results of node classification (*i.e.,* Macro-F1 and Micro-F1) in Table 1 and Appendix E, and reporting the results of similarity search (*i.e.,* Sim@5 and Sim@10) in Appendix E. Obviously, our method achieves superior performance on both node classification and similarity search tasks.

First, for the node classification task, the proposed method always outperforms the comparison methods by large margins. For example, the proposed method on average, improves by 2.1%, compared to the best SHGL method (*i.e.,* DMG), on four heterogeneous graph datasets. The reason can be attributed to the fact that the proposed method extracts both homophily and heterogeneity in the heterogeneous graph, thus introducing more task-related information to improve the effectiveness of the classification task. Second, for the similarity search task, the proposed method also obtains promising improvements. For example, the proposed method on average, improves by 1.8%, compared to the best SHGL method (*i.e.,* DMG), on four heterogeneous graph datasets. This demonstrates the superiority of the proposed method, which captures the homophily in the heterogeneous graph, enforcing the representations to have the grouping effect and thus increasing the similarity of nodes within the same class. As a result, the effectiveness of the proposed method is verified on the heterogeneous graph datasets in terms of different downstream tasks.

### 3.2.2 EFFECTIVENESS ON THE HOMOGENEOUS GRAPH

We further evaluate the effectiveness of the proposed method on the homogeneous graph datasets by reporting the results of node classification (*i.e.,* Macro-F1 and Micro-F1) in Table 2. We can observe that the proposed method achieves competitive results on the homogeneous graph datasets.

First, compared to the semi-supervised baselines (*i.e.,* GCN and GAT), the proposed method always achieves the best results. For example, the proposed method on average, improves by 1.1%, compared to the best semi-supervised method (*i.e.,* GCN), on four homogeneous graph datasets. Second, compared to the self-supervised methods, the proposed method also achieves superior performance.

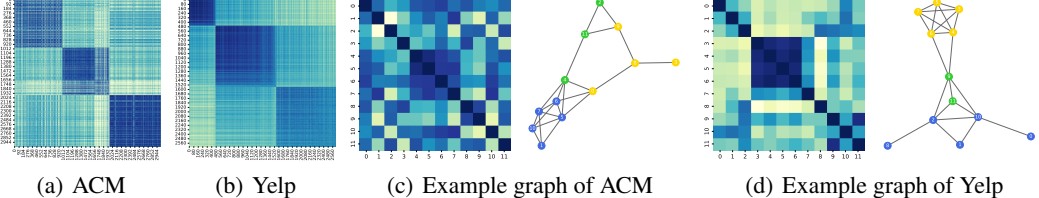

| (a) ACM | (b) Yelp | (c) Example graph of ACM | (d) Example graph of Yelp |

Figure 3: (a) and (b) indicate node correlation maps of ACM and Yelp datasets reordered by node labels. (c) and (d) indicate node correlation maps and corresponding visualizations (top 30% values in the correlation map are visualized as edges) of example graphs of ACM and Yelp datasets.

For example, the proposed method outperforms the best self-supervised method (*i.e.,* COSTA), on almost all homogeneous graph datasets. This indicates that the proposed method extracts the consistent information between the original graph and the graph with homophily, as well as preserves the specific information within each of them to benefit downstream tasks. As a result, the effectiveness of the proposed method is further verified on the homogeneous graph datasets.

### 3.2.3 VISUALIZATION AND CASE STUDY

**Visualization of Grouping Effect.** To verify the grouping effect of homophilous representations, we visualize the node correlation maps of ACM and Yelp datasets in Figures 3(a) and 3(b), where rows and columns are reordered by node labels. In the correlation map, the darker a pixel, the higher the correlation between nodes. In Figures 3(a) and 3(b), the correlation maps exhibit a block diagonal structure where the nodes of each block belong to the same class. This indicates that if two nodes belong to the same class, then the correlation of their representations will be high, *i.e.,* their representations are expected to be aligned. This verifies the grouping effect of the homophilous representations, which capture the homophily in the heterogeneous graph.

**Visualization of Example Graph.** To further verify that the homophilous representations indeed capture the homophily in the heterogeneous graph, we sample example graphs from ACM and Yelp datasets and visualize the correlation maps among sampled nodes in Figures 3(c) and 3(d). Moreover, we visualize the top 30% values in the correlation maps as edges between nodes to make them more intuitive. In Figures 3(c) and 3(d), we have the observations as follows. First, two nodes (*e.g.,* node 2 and node 11) within the same class do have high correlation values, while two nodes (*e.g.,* node 2 and node 1) from different classes do have low correlation values. Second, the visualized edges in Figures 3(c) and 3(d) indeed show a high homophily rate (*e.g.,* 73% in the example graph of the Yelp dataset). This further verifies that the homophilous representations indeed capture the homophily in the heterogeneous graph without pre-defined meta-paths.

## 4 CONCLUSION

In this paper, we proposed a self-supervised heterogeneous graph learning framework to capture both homophily and heterogeneity in the heterogeneous graph without pre-defined meta-paths. To do this, we proposed to learn a self-expressive matrix adaptively and employ the heterogeneous encoder to obtain homophilous and heterogeneous representations for capturing homophily and heterogeneity in the heterogeneous graph, respectively. We further designed the consistency loss and the specificity loss to extract the consistent information between homophilous representations and heterogeneous representations and to maintain their specific information in different latent spaces, respectively. Theoretical analysis indicates that the homophilous representations capture the homophily in the heterogeneous graph. In addition, the fused representations are provable to contain more task-related information than the representations with homophily or heterogeneity only, thus benefiting downstream tasks. Extensive experimental results demonstrate the effectiveness of the proposed method on both homogeneous and heterogeneous graph datasets in terms of different downstream tasks. We discuss potential limitations and future work in Appendix F.

## 5   ACKNOWLEDGMENTS

This work was supported in part by the National Key Research and Development Program of China under Grant 2022YFA1004100, in part by the Medico-Engineering Cooperation Funds from University of Electronic Science and Technology of China under Grant ZYGX2022YGRH009 and Grant ZYGX2022YGRH014, in part by the National Natural Science Foundation of China under Grant 62276052.

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

# A    RELATED WORK

This section briefly reviews topics related to this work, including self-supervised learning in Section A.1, heterogeneous graph learning in Section A.2, and self-attention mechanism in Section A.3.

## A.1    SELF-SUPERVISED LEARNING

Self-supervised learning (SSL) has emerged as a promising approach to address the challenge of acquiring labeled data for training deep neural networks. Unlike traditional supervised learning, where labeled data is required, SSL leverages the abundance of unlabeled data to learn meaningful representations. Therefore, SSL has shown strength in various domains such as computer vision (Grill et al., 2020; Yang et al., 2022; Zhang et al., 2023a), natural language processing (Devlin et al., 2019; Klein & Nabi, 2020; Lan et al., 2020), and graph representation learning (Jin et al., 2023; Liu et al., 2023b; Sun et al., 2023a; Liang et al., 2024).

Previous SSL methods generally conduct contrastive learning between the positive pairs and negative pairs to maximize the mutual information between the local representations and their related representations. For example, Deep InfoMax (Hjelm et al., 2019) conducts contrastive learning by maximizing the mutual information between a local patch and its global context. CPC (Oord et al., 2018) achieves great results on the speech recognition task by maximizing the mutual information between the local audio representations and its global audio representations. SimCLR (Chen et al., 2020) conducts the contrastive learning between the original view and the augmented view, and argues that data augmentation plays an important role in contrastive learning. (Sun et al., 2023b) proposes a novel hypergraph neural network to learn influence flowing under social criteria with dual contrastive loss. To remove the need for negative samples in contrastive SSL methods, recent works propose to conduct self-supervised learning without negative samples. For instance, BYOL (Grill et al., 2020) leverages a target network to generate target representations for an augmented view, and the online network aims to match its own predictions with the target representations. Barlow Twins (Zbontar et al., 2021) propose to capture the invariant information between the original view and the augmented view as well as decorrelate different representation dimensions. E-SSL (Dangovski et al., 2022) encourages the equivariance to some transformations, while maintaining the invariance to other transformations, thus improving the semantic quality of representations.

## A.2    HETEROGENEOUS GRAPH LEARNING

Heterogeneous graph learning methods aim to explore the latent patterns in the heterogeneous graph and have been applied to various practical applications. Considering the cost of obtaining node labels, self-supervised heterogeneous graph learning has drawn much attention (Liang et al., 2022; Tian et al., 2022; Liu et al., 2023a; Yang et al., 2023; Wang et al., 2023). Existing SHGL methods generally employ meta-paths to extract distinct semantic relationships between nodes of the same type while ignoring other node types.

In light of such meta-path preprocessing, existing SHGL methods can be broadly categorized into two groups, *i.e.,* intra-path learning methods and inter-path learning methods. Intra-path learning methods focus on capturing the global properties within each meta-path-based graph to enhance the quality of node representations. For instance, DMGI (Park et al., 2020) and HDMI (Jing et al., 2021) utilize contrastive learning techniques to establish connections between node representations and graph summaries, compelling the incorporation of global properties within the node representations. Inter-path learning methods aim to capture the associations as well as invariant information across different meta-path-based graphs. For example, HGCML (Wang et al., 2023) and CPIM (Mo et al., 2023b) propose to maximize the mutual information between node representations from different meta-path-based graphs. Although existing SHGL methods have shown great potential and achieved impressive results in various tasks, they generally require pre-defined meta-paths, which induce substantial expert knowledge and expensive costs. Moreover, many previous SHGL methods ignore the heterogeneity in the heterogeneous graph to lose task-related information. Even though a few works (Wang et al., 2021) explore the neighbors of nodes from different types in the network schema, they directly enforce the homophily of meta-paths and the heterogeneity of the network schema to align with each other by contrastive learning. As a result, they still need pre-defined meta-paths and may lose distinct information in the homophily and heterogeneity to weaken downstream tasks.

### A.3 Self-attention Mechanism

Self-attention is a mechanism that allows a model to focus on all samples within a sequence or set and learn relationships between them. It enables the model to assign varying levels of importance or attention to different samples based on their relevance to each other (Velickovic et al., 2018; Devlin et al., 2019; Liang et al., 2023; Zhang et al., 2023b). As a result, the self-attention mechanism has achieved promising results in natural language processing tasks (*e.g.,* text classification (Galassi et al., 2020), and machine translation (Stahlberg, 2020)) and computer vision tasks (*e.g.,* image recognition and object detection (Zhao et al., 2021b)).

Actually, the self-attention mechanism is closely related to the notion of the self-expressive matrix in the proposed method, wherein each sample is expressed as linear combinations of all other samples, thus capturing the global correlation among all samples. However, there are also some major differences between them. First, the self-expressive matrix is not restricted to be non-negative, allowing for both positive and negative attention, while the self-attention mechanism is restricted to be non-negative. Second, the self-attention mechanism is defined as a function of the tokens that are parameterized by learnable weights. Instead, the self-expressive matrix in the proposed method is the closed-form solution derived from an unsupervised objective function without parameters. Third, the self-expressive matrix in the proposed method is regularized by the feature distance matrix and is encouraged to focus on the most relevant neighbors of each sample instead of arbitrary neighbors in the self-attention mechanism. Fourth, the time complexity of the closed-form solution of the self-expressive matrix can be reduced to scale linearly with the sample size by the matrix identity transformation, while the self-attention mechanism requires quadratic time complexity with the sample size. Even though part of works (Wang et al., 2020; Choromanski et al., 2020) propose to reduce the time complexity of the self-attention mechanism, they sacrifice the model performance by the approximation transformation. As a result, the self-expressive matrix is more general and flexible than the self-attention mechanism, and therefore we employ it to capture the homophily in the heterogeneous graph.

## B    Algorithm and Complexity Analysis

This section provides the pseudo-code of the proposed method in Section B.1, and the complexity analysis of our method in Section B.2.

### B.1    Algorithm

---
**Algorithm 1** The pseudo-code of the proposed method.

---
**Input:** Heterogeneous graph $\mathbf{G} = (\mathcal{V}, \mathcal{E}, \mathbf{X}, \mathcal{T}, \mathcal{R})$, non-negative parameters $\alpha$, $\beta$, $\gamma$, $\eta$, and $\lambda$;
**Output:** Encoders $g_\phi$, $f_\theta$;
 1: Initialize parameters;
 2: **while** not converge **do**
 3:     Obtain node representations with encoder $g_\phi$;
 4:     Obtain the closed-form solution of self-expressive matrix $\mathbf{S}^*$ by Eq. (4);
 5:     Obtain homophilous representations $\mathbf{Z}$ by Eq. (5);
 6:     Transform the matrix and reorder the matrix multiplication to reduce the time complexity by Eq. (6);
 7:     Obtain heterogeneous representations with encoder $f_\theta$;
 8:     Project homophilous representations and heterogeneous representations into a latent space to obtain $\mathbf{P}$ and $\widetilde{\mathbf{P}}$;
 9:     Conduct the consistency loss between $\mathbf{P}$ and $\widetilde{\mathbf{P}}$ by Eq. (10);
10:     Project homophilous representations and heterogeneous representations into another latent space to obtain $\mathbf{Q}$ and $\widetilde{\mathbf{Q}}$;
11:     Transform $\mathbf{Q}$ with a transformation head to obtain $\mathbf{Q}'$;
12:     Conduct the specificity loss between $\mathbf{Q}'$ and $\widetilde{\mathbf{Q}}$ by Eq. (11);
13:     Compute the objective function $\mathcal{J}$ by Eq. (12);
14:     Back-propagate $\mathcal{J}$ to update model weights;
15: **end while**

---

## B.2 COMPLEXITY ANALYSIS

Based on the Algorithm 1 above, we then analyze the time complexity of the proposed method. We define $N$ and $d$ as the number of nodes, and dimension of representations, respectively. According to Eq. (6), we can obtain the homophilous representations $\mathbf{Z}$ as follows.

$$\mathbf{Z} = \mathbf{H}^{(l+1)}(\mathbf{H}^{(l+1)})^T\mathbf{B} - \frac{\alpha}{2}\mathbf{DB}, \qquad (14)$$

where $\mathbf{H}^{(l+1)}, \mathbf{B} \in \mathbb{R}^{N \times d}$, $\mathbf{B} = \frac{1}{\beta}\mathbf{H}^{(l+1)} - \frac{1}{\beta^2}\mathbf{H}^{(l+1)}(\mathbf{I}_d + \frac{1}{\beta}(\mathbf{H}^{(l+1)})^T\mathbf{H}^{(l+1)})^{-1}(\mathbf{H}^{(l+1)})^T\mathbf{H}^{(l+1)}$, and $\mathbf{I}_d \in \mathbb{R}^{d \times d}$ indicates the identity matrix. Therefore, we first calculate $\mathbf{B}$ and calculate the second term of it from right to left. Specifically, the matrix inversion is conducted on a matrix $\mathbb{R}^{d \times d}$, whose time complexity is $\mathcal{O}(d^3)$. Therefore, the overall time complexity of calculating $\mathbf{B}$ is $\mathcal{O}(Nd^2 + d^3)$. After that, we obtain the homophilous representations with Eq. (14). In Eq. (14), the time complexity to calculate $\mathbf{DB}$ is $\mathcal{O}(kd)$, where $k$ indicates the nonzero entries of the sparse feature distance matrix $\mathbf{D}$ and $k \ll N^2$. In addition, the time complexity of calculating $\mathbf{H}^{(l+1)}(\mathbf{H}^{(l+1)})^T\mathbf{B}$ is $\mathcal{O}(Nd^2)$. Therefore, the time complexity of calculating homophilous representations $\mathbf{Z}$ is $\mathcal{O}(Nd^2 + d^3 + kd)$. Moreover, the time complexity of Eq. (10) is $\mathcal{O}(Nd^2)$, and the time complexity of Eq. (11) is $\mathcal{O}(N)$. Therefore, the overall complexity of our algorithm is $\mathcal{O}(Nd^2 + d^3 + kd)$ in each epoch, which is scaled linearly with the sample size.

## C DERIVATION PROCESS AND PROOFS OF THEOREMS

This section provides definition, detailed derivation process, and proofs of Theorems in Section 2, including the definition of homophily mining, homophily ratio, and heterogeneity mining in Section C.1, the derivation process of the closed-form solution in Section C.2, matrix transformation in Section C.3, proof of Theorem 2.2 in Section C.4, proof of Theorem 2.3 in Section C.5, and the Corollary as well as its corresponding proof in Section C.6.

## C.1 DEFINITIONS

According to previous works (Zhu et al., 2020a; Ma et al., 2021), we have the definition as follows.

**Definition C.1.** *(Homophily mining) Given a graph $\mathcal{G} = \{\mathcal{V}, \mathcal{E}\}$, the homophily mining is the connectivity utilization and information aggregation among nodes within the same class. Formally, we have*

$$\mathbf{m}_i^{(l)} = \mathrm{MES}^{(l)}\left(\left\{\mathbf{h}_j^{(l-1)} \mid (v_i, v_j) \in \mathcal{E}, y_i = y_j\right\}\right), \qquad (15)$$

*where $l$ indicates the $l$-th layer, MES indicates the message passing function, $\mathbf{m}_i$ and $\mathbf{h}_j$ indicate node representations of node $v_i$ and node $v_j$, $y_i$ and $y_j$ indicate labels of node $v_i$ and node $v_j$.*

**Definition C.2.** *(Homophily ratio) Given a graph $\mathcal{G} = \{\mathcal{V}, \mathcal{E}\}$, the edge homophily ratio $h$ is defined as the fraction of edges that connect nodes within the same class. Formally, we have*

$$h = \frac{1}{|\mathcal{E}|} \sum_{(v_i, v_j) \in \mathcal{E}} \mathbb{1}\left(y_i = y_j\right), \qquad (16)$$

*where $|\mathcal{E}|$ is the number of edges in the graph and $\mathbb{1}(\cdot)$ is the indicator function.*

**Definition C.3.** *(Heterogeneity mining) Given a heterogeneous graph $\mathbf{G} = (\mathcal{V}, \mathcal{E}, \mathbf{X}, \mathcal{T}, \mathcal{R})$, the heterogeneity mining is the connectivity utilization and information aggregation among nodes among nodes from different types. Formally, we have*

$$\mathbf{m}_i^{(l)} = \mathrm{MES}^{(l)}\left(\left\{\mathbf{h}_j^{(l-1)} \mid (v_i, v_j) \in \mathcal{E}_r, v_i \in \mathcal{V}_\tau, v_j \in \mathcal{V}_{\tau'}\right\}\right), \qquad (17)$$

*where $l$ indicates the $l$-th layer, MES indicates the message passing function, $\mathbf{m}_i$ and $\mathbf{h}_j$ indicate node representations of node $v_i$ and node $v_j$, $\mathcal{E}_r$ indicates the $r$-th type of edges, $\mathcal{V}_\tau$ and $\mathcal{V}_{\tau'}$ indicate the $\tau$-th and $\tau'$-th type of nodes, and $\tau \neq \tau'$.*

## C.2 CLOSED-FORM SOLUTION

Given the objective function in Eq. (3), we let

$$
\begin{aligned}
J &= \|\mathbf{H}^{(l+1)} - \mathbf{S}\mathbf{H}^{(l+1)}\|_F^2 + \alpha \sum_{i,j=1}^{N} \|\mathbf{x}_i - \mathbf{x}_j\|_2^2 s_{ij} + \beta \sum_{i,j=1}^{N} s_{ij}^2 \\
&= \mathrm{Tr}((\mathbf{H}^{(l+1)} - \mathbf{S}\mathbf{H}^{(l+1)})^T(\mathbf{H}^{(l+1)} - \mathbf{S}\mathbf{H}^{(l+1)})) + \alpha \mathrm{Tr}(\mathbf{S}^T\mathbf{D}) + \beta \mathrm{Tr}(\mathbf{S}^T\mathbf{S}),
\end{aligned}
\tag{18}
$$

where $\mathrm{Tr}(\cdot)$ indicates the trace of matrix. Then we have

$$
\frac{\partial J}{\partial \mathbf{S}} = -2\mathbf{H}^{(l+1)}(\mathbf{H}^{(l+1)})^T + 2\mathbf{S}\mathbf{H}^{(l+1)}(\mathbf{H}^{(l+1)})^T + \alpha\mathbf{D} + 2\beta\mathbf{S}.
\tag{19}
$$

Let Eq. (19) equal to 0, we can obtain the closed-form solution $\mathbf{S}^*$ of Eq. (3), *i.e.,*

$$
\mathbf{S}^* = (\mathbf{H}^{(l+1)}(\mathbf{H}^{(l+1)})^T - \frac{\alpha}{2}\mathbf{D})(\mathbf{H}^{(l+1)}(\mathbf{H}^{(l+1)})^T + \beta\mathbf{I}_N)^{-1}.
\tag{20}
$$

## C.3 MATRIX TRANSFORMATION

Given four matrix, *i.e.,* $\mathbf{A} \in \mathbb{R}^{n \times n}$, $\mathbf{U} \in \mathbb{R}^{n \times k}$, $\mathbf{C} \in \mathbb{R}^{k \times k}$, and $\mathbf{V} \in \mathbb{R}^{k \times n}$, where $n$, $k$ are dimensions of these matrix, according to the Woodbury identity matrix transformation (Woodbury, 1950), we have

$$
(\mathbf{A} + \mathbf{U}\mathbf{C}\mathbf{V})^{-1} = \mathbf{A}^{-1} - \mathbf{A}^{-1}\mathbf{U}\left(\mathbf{C}^{-1} + \mathbf{V}\mathbf{A}^{-1}\mathbf{U}\right)^{-1}\mathbf{V}\mathbf{A}^{-1}.
\tag{21}
$$

Without loss of generality, the matrix $\mathbf{A}$ and $\mathbf{C}$ can be replaced with the identity matrix, therefore, we further have

$$
(\mathbf{I} + \mathbf{U}\mathbf{V})^{-1} = \mathbf{I} - \mathbf{U}(\mathbf{I} + \mathbf{V}\mathbf{U})^{-1}\mathbf{V}.
\tag{22}
$$

Based on Eq. (22), we can transform $(\mathbf{H}^{(l+1)}(\mathbf{H}^{(l+1)})^T + \beta\mathbf{I}_N)^{-1}$ in Eq. (4) as:

$$
(\mathbf{H}^{(l+1)}(\mathbf{H}^{(l+1)})^T + \beta\mathbf{I}_N)^{-1} = \frac{1}{\beta}\mathbf{I} - \frac{1}{\beta^2}\mathbf{H}^{(l+1)}(\mathbf{I}_d + \frac{1}{\beta}(\mathbf{H}^{(l+1)})^T\mathbf{H}^{(l+1)})^{-1}(\mathbf{H}^{(l+1)})^T.
\tag{23}
$$

Therefore, with Eq. (4) and Eq. (23), we can transform Eq. (5) as

$$
\begin{aligned}
\mathbf{Z} &= \mathbf{S}^*\mathbf{H}^{(l+1)} \\
&= (\mathbf{H}^{(l+1)}(\mathbf{H}^{(l+1)})^T - \frac{\alpha}{2}\mathbf{D})(\mathbf{H}^{(l+1)}(\mathbf{H}^{(l+1)})^T + \beta\mathbf{I}_N)^{-1}\mathbf{H}^{(l+1)} \\
&= (\mathbf{H}^{(l+1)}(\mathbf{H}^{(l+1)})^T - \frac{\alpha}{2}\mathbf{D})(\frac{1}{\beta}\mathbf{I} - \frac{1}{\beta^2}\mathbf{H}^{(l+1)}(\mathbf{I}_d + \frac{1}{\beta}(\mathbf{H}^{(l+1)})^T\mathbf{H}^{(l+1)})^{-1}(\mathbf{H}^{(l+1)})^T)\mathbf{H}^{(l+1)}.
\end{aligned}
\tag{24}
$$

Then we replace $(\frac{1}{\beta}\mathbf{I} - \frac{1}{\beta^2}\mathbf{H}^{(l+1)}(\mathbf{I}_d + \frac{1}{\beta}(\mathbf{H}^{(l+1)})^T\mathbf{H}^{(l+1)})^{-1}(\mathbf{H}^{(l+1)})^T)\mathbf{H}^{(l+1)}$ with $\mathbf{B}$ and obtain

$$
\begin{aligned}
\mathbf{Z} &= (\mathbf{H}^{(l+1)}(\mathbf{H}^{(l+1)})^T - \frac{\alpha}{2}\mathbf{D})\mathbf{B} \\
&= \mathbf{H}^{(l+1)}(\mathbf{H}^{(l+1)})^T\mathbf{B} - \frac{\alpha}{2}\mathbf{D}\mathbf{B}.
\end{aligned}
\tag{25}
$$

This is exactly the Eq. (6).

## C.4 PROOF OF THEOREM 2.2

**Theorem C.4.** (Restating Theorem 2.2 in the main text). *Both self-expressive matrix $\mathbf{S}^* \in \mathbb{R}^{N \times N}$ and homophilous representations $\mathbf{Z} \in \mathbb{R}^{N \times d}$ have the grouping effect for any two nodes $v_i$ and $v_j$ that hold the condition $v_i \to v_j$, i.e.,*

$$
v_i \to v_j \Rightarrow \left|s_{ip}^* - s_{jp}^*\right| \to 0, \text{ and } |z_{iq} - z_{jq}| \to 0, \forall 1 \le q \le d, 1 \le i, j, p \le N.
\tag{26}
$$

*Proof.* For $1 \leq i \leq N$, we denote $\mathbf{s}_i$ as the $i$-th row of the coefficient matrix $\mathbf{S}$, and denote $\mathbf{d}_i$ as the $i$-th row of the feature distance matrix $\mathbf{D}$. According to the Eq. (3), we let $J(\mathbf{s}_i) = \|\mathbf{h}_i^{(l+1)} - \mathbf{s}_i \mathbf{H}^{(l+1)}\|_2^2 + \alpha \mathbf{d}_i(\mathbf{s}_i)^T + \beta \|\mathbf{s}_i\|_2^2$. With the closed-form solution in Eq. (4), we have $\frac{\partial J}{\partial \mathbf{S}_{ip}}\big|_{\mathbf{s}_i = \mathbf{s}_i^*} = 0, \forall 1 \leq p \leq N$. Therefore, we have $-(\mathbf{h}_i^{(l+1)} - \mathbf{s}_i^* \mathbf{H}^{(l+1)})(\mathbf{h}_p^{(l+1)})^T + \frac{\alpha}{2} d_{ip} + \beta s_{ip}^* = 0$. We further have

$$s_{ip}^* = \frac{(\mathbf{h}_i^{(l+1)} - \mathbf{s}_i^* \mathbf{H}^{(l+1)})(\mathbf{h}_p^{(l+1)})^T - \frac{\alpha}{2} d_{ip}}{\beta}. \tag{27}$$

Based on Eq. (27), we can obtain

$$s_{ip}^* - s_{jp}^* = \frac{(\mathbf{h}_i^{(l+1)} - \mathbf{h}_j^{(l+1)})(\mathbf{h}_p^{(l+1)})^T - (\mathbf{s}_i^* - \mathbf{s}_j^*)\mathbf{H}^{(l+1)}(\mathbf{h}_p^{(l+1)})^T - \frac{\alpha}{2}(d_{ip} - d_{jp})}{\beta}. \tag{28}$$

Moreover, according to Eq. (4), we have

$$\mathbf{s}_i^* = (\mathbf{h}_i^{(l+1)}(\mathbf{H}^{(l+1)})^T - \frac{\alpha}{2}\mathbf{d}_i)(\mathbf{H}^{(l+1)}(\mathbf{H}^{(l+1)})^T + \beta \mathbf{I}_N)^{-1}. \tag{29}$$

Therefore,

$$\mathbf{s}_i^* - \mathbf{s}_j^* = ((\mathbf{h}_i^{(l+1)} - \mathbf{h}_j^{(l+1)})(\mathbf{H}^{(l+1)})^T - \frac{\alpha}{2}(\mathbf{d}_i - \mathbf{d}_j))(\mathbf{H}^{(l+1)}(\mathbf{H}^{(l+1)})^T + \beta \mathbf{I}_N)^{-1}, \tag{30}$$

Replace $(\mathbf{H}^{(l+1)}(\mathbf{H}^{(l+1)})^T + \beta \mathbf{I}_N)^{-1}\mathbf{H}^{(l+1)}(\mathbf{h}_p^{(l+1)})^T$ with $\mathbf{M}$, then we can rewrite Eq. (28) as:

$$\begin{aligned}
s_{ip}^* - s_{jp}^* &= \frac{(\mathbf{h}_i^{(l+1)} - \mathbf{h}_j^{(l+1)})(\mathbf{h}_p^{(l+1)})^T}{\beta} - \frac{((\mathbf{h}_i^{(l+1)} - \mathbf{h}_j^{(l+1)})(\mathbf{H}^{(l+1)})^T - \frac{\alpha}{2}(\mathbf{d}_i - \mathbf{d}_j))\mathbf{M}}{\beta} \\
&\quad - \frac{(\frac{\alpha}{2}(d_{ip} - d_{jp}))}{\beta} \\
&= \frac{(\mathbf{h}_i^{(l+1)} - \mathbf{h}_j^{(l+1)})((\mathbf{h}_p^{(l+1)})^T - (\mathbf{H}^{(l+1)})^T\mathbf{M}))}{\beta} + \frac{\frac{\alpha}{2}(\mathbf{d}_i - \mathbf{d}_j)\mathbf{M}}{\beta} - \frac{\frac{\alpha}{2}(d_{ip} - d_{jp})}{\beta}.
\end{aligned} \tag{31}$$

Therefore, we have

$$\begin{aligned}
|s_{ip}^* - s_{jp}^*| &\leq \frac{\|(\mathbf{h}_i^{(l+1)} - \mathbf{h}_j^{(l+1)})\|_2 \|(\mathbf{h}_p^{(l+1)})^T - (\mathbf{H}^{(l+1)})^T\mathbf{M}\|_2}{\beta} + \frac{\frac{\alpha}{2}\|(\mathbf{d}_i - \mathbf{d}_j)\|_2 \|\mathbf{M}\|_2}{\beta} \\
&\quad - \frac{\frac{\alpha}{2}|d_{ip} - d_{jp}|}{\beta}.
\end{aligned} \tag{32}$$

If the condition that $v_i \to v_j$ holds, *i.e.*, $\|\mathbf{x}_i - \mathbf{x}_j\|_2 \to 0$ hold, then we have $|d_{ip} - d_{jp}| \to 0, \forall 1 \leq p \leq N$, *i.e.*, $\|\mathbf{d}_i - \mathbf{d}_j\|_2 \to 0$. Moreover, we can obtain $\|(\mathbf{h}_i^{(l+1)} - \mathbf{h}_j^{(l+1)})\|_2 \to 0$ according to Eq. (1). As a result, the R.H.S. of Eq. (32) close to 0, and $|s_{ip}^* - s_{jp}^*| \to 0$ holds, thus $\mathbf{S}^*$ has the grouping effect. In addition, for any $1 \leq q \leq d$, denote $\mathbf{h}^q$ as the $q$-th column of $\mathbf{H}^{(l+1)}$, we have $z_{iq} - z_{jq} = \mathbf{s}_i^* \mathbf{h}^q - \mathbf{s}_j^* \mathbf{h}^q = (\mathbf{s}_i^* - \mathbf{s}_j^*)\mathbf{h}^q$. Therefore, we further have $|z_{iq} - z_{jq}| \leq \|\mathbf{s}_i^* - \mathbf{s}_j^*\|_2 \|\mathbf{h}^q\|_2$. If the condition that $v_i \to v_j$ holds, we obtain $\|\mathbf{s}_i^* - \mathbf{s}_j^*\|_2 \to 0$ according to Eq. (30). That is, if $v_i \to v_j$ holds, then $|z_{iq} - z_{jq}| \to 0$ holds with $\forall 1 \leq q \leq d$, thus homophilous representations $\mathbf{Z}$ have the grouping effect. Therefore, we complete the proof. $\square$

## C.5 PROOF OF THEOREM 2.3

In the following proofs, for random variables A, B, C, we use $I(A, B)$ to represent the mutual information between A and B, and we use $I(A, B|C)$ to represent conditional mutual information of A and B on a given C, use $H(A)$ for the entropy, and $H(A|B)$ for the conditional entropy. We first list some properties of mutual information and entropy that will be used in the proofs.

- **Property 1.** Relationship between the mutual information and entropy:

$$I(A, B) = H(A) - H(A \mid B). \tag{33}$$

- **Property 2.** Relationship between the conditional mutual information and entropy:

$$I(A, B \mid C) = H(A \mid C) - H(A \mid B, C). \tag{34}$$

- **Property 3.** Non-negativity of mutual information:

$$I(A, B) \geq 0, I(A, B \mid C) \geq 0. \tag{35}$$

- **Property 4.** Relationship between the conditional entropy and entropy:

$$H(A \mid B) = H(A, B) - H(B). \tag{36}$$

**Theorem C.5.** (Restating Theorem 2.3 in the main text). *For any downstream task $T$, the representations with both homophily and heterogeneity (e.g., $\hat{\mathbf{Z}}$) contain more task-related information than the representations with only homophily (e.g., $\mathbf{Z}$) or with only heterogeneity (e.g., $\widetilde{\mathbf{Z}}$), i.e.,*

$$I(\hat{\mathbf{Z}}, T) \geq \max(I(\mathbf{Z}, T), I(\widetilde{\mathbf{Z}}, T)), \tag{37}$$

where $I(\cdot, \cdot)$ indicates the mutual information.

*Proof.* Given the fused representations $\hat{\mathbf{Z}}$ that contain both homophily and heterogeneity, we have

$$H(\hat{\mathbf{Z}}) = H(\mathbf{Z}|\widetilde{\mathbf{Z}}) + H(\widetilde{\mathbf{Z}}|\mathbf{Z}) + I(\widetilde{\mathbf{Z}}, \mathbf{Z}), \tag{38}$$

where $H(\mathbf{Z}|\widetilde{\mathbf{Z}})$ and $H(\widetilde{\mathbf{Z}}|\mathbf{Z})$ indicate the specific information of $\mathbf{Z}$ and $\widetilde{\mathbf{Z}}$, respectively, and $I(\widetilde{\mathbf{Z}}, \mathbf{Z})$ indicates the consistent information between $\mathbf{Z}$ and $\widetilde{\mathbf{Z}}$. According to Properties 1 and 4, we have

$$\begin{aligned} H(\hat{\mathbf{Z}}) &= H(\mathbf{Z}|\widetilde{\mathbf{Z}}) + H(\widetilde{\mathbf{Z}}|\mathbf{Z}) + I(\mathbf{Z}, \widetilde{\mathbf{Z}}) \\ &= H(\mathbf{Z}|\widetilde{\mathbf{Z}}) + H(\widetilde{\mathbf{Z}}|\mathbf{Z}) + H(\mathbf{Z}) - H(\mathbf{Z}|\widetilde{\mathbf{Z}}) \\ &= H(\mathbf{Z}|\widetilde{\mathbf{Z}}) + H(\widetilde{\mathbf{Z}}|\mathbf{Z}) + H(\mathbf{Z}, \widetilde{\mathbf{Z}}) - H(\widetilde{\mathbf{Z}}|\mathbf{Z}) - H(\mathbf{Z}|\widetilde{\mathbf{Z}}) \\ &= H(\mathbf{Z}, \widetilde{\mathbf{Z}}). \end{aligned} \tag{39}$$

Therefore, for any downstream task $T$, we further have

$$H(\hat{\mathbf{Z}}, T) = H(\mathbf{Z}, \widetilde{\mathbf{Z}}, T). \tag{40}$$

To prove $I(\hat{\mathbf{Z}}, T) \geq \max(I(\mathbf{Z}, T), I(\widetilde{\mathbf{Z}}, T))$, we first prove $I(\hat{\mathbf{Z}}, T) \geq I(\mathbf{Z}, T)$. Then based on Eq. (39), Eq. (40), Property 1, and Property 4, we can transform $I(\hat{\mathbf{Z}}, T)$ as follows.

$$\begin{aligned} I(\hat{\mathbf{Z}}, T) &= H(\hat{\mathbf{Z}}) - H(\hat{\mathbf{Z}}|T) \\ &= H(\hat{\mathbf{Z}}) - H(\hat{\mathbf{Z}}, T) + H(T) \\ &= H(\mathbf{Z}, \widetilde{\mathbf{Z}}) - H(\mathbf{Z}, \widetilde{\mathbf{Z}}, T) + H(T). \end{aligned} \tag{41}$$

Moreover, based on Properties 1 and 2, we have

$$I(\mathbf{Z}, T) = H(\mathbf{Z}) - H(\mathbf{Z}|T). \tag{42}$$

$$\begin{aligned} I(\widetilde{\mathbf{Z}}, T|\mathbf{Z}) &= H(\widetilde{\mathbf{Z}}|\mathbf{Z}) - H(\widetilde{\mathbf{Z}}|\mathbf{Z}, T) \\ &= H(\mathbf{Z}, \widetilde{\mathbf{Z}}) - H(\mathbf{Z}) - H(\widetilde{\mathbf{Z}}|\mathbf{Z}, T). \end{aligned} \tag{43}$$

Then with Eq. (42), Eq. (43) and Property 4, we can obtain

$$\begin{aligned} I(\mathbf{Z}, T) + I(\widetilde{\mathbf{Z}}, T|\mathbf{Z}) &= H(\mathbf{Z}) - H(\mathbf{Z}|T) + H(\mathbf{Z}, \widetilde{\mathbf{Z}}) - H(\mathbf{Z}) - H(\widetilde{\mathbf{Z}}|\mathbf{Z}, T) \\ &= H(\mathbf{Z}, \widetilde{\mathbf{Z}}) - H(\mathbf{Z}|T) - H(\widetilde{\mathbf{Z}}|\mathbf{Z}, T) \\ &= H(\mathbf{Z}, \widetilde{\mathbf{Z}}) - H(\mathbf{Z}, T) + H(T) - H(\widetilde{\mathbf{Z}}|\mathbf{Z}, T) \\ &= H(\mathbf{Z}, \widetilde{\mathbf{Z}}) - H(\mathbf{Z}, T) + H(T) - H(\widetilde{\mathbf{Z}}, \mathbf{Z}, T) + H(\mathbf{Z}, T) \\ &= H(\mathbf{Z}, \widetilde{\mathbf{Z}}) + H(T) - H(\widetilde{\mathbf{Z}}, \mathbf{Z}, T). \end{aligned} \tag{44}$$

According to Eq. (41) and Eq. (44), we have

$$I(\hat{\mathbf{Z}}, T) = I(\mathbf{Z}, T) + I(\widetilde{\mathbf{Z}}, T|\mathbf{Z}). \tag{45}$$

Based on Property 3, we have $I(\widetilde{\mathbf{Z}}, T|\mathbf{Z}) \geq 0$, the we can get

$$I(\hat{\mathbf{Z}}, T) \geq I(\mathbf{Z}, T). \tag{46}$$

Similarly, we can also obtain

$$I(\hat{\mathbf{Z}}, T) \geq I(\widetilde{\mathbf{Z}}, T). \tag{47}$$

Therefore, $I(\hat{\mathbf{Z}}, T) \geq \max(I(\mathbf{Z}, T), I(\widetilde{\mathbf{Z}}, T))$ and we complete the proof. $\square$

## C.6 COROLLARY C.6

To further verify Theorem 2.3, we take the classification as an example downstream task $T$, then employ the Bayes error rate (Feder & Merhav, 1994), which is the lowest achievable error when learning an arbitrary classifier from the representation to infer the labels. Specifically, let $P_e$ be the Bayes error rate of arbitrary learned representations $\hat{\mathbf{Z}}$ and $\widehat{T}$ as the prediction for $T$ from the classifier. Thus, we have $P_e = \mathbb{E}_{\hat{\mathbf{Z}} \sim P_{\hat{\mathbf{Z}}}}[1 - \max_{t \in T} P(\hat{T} = t \mid \hat{\mathbf{Z}})]$. To prevent overflow, we further define $\bar{P}_e = \text{Th}(P_e)$, where $\text{Th}(x) = \min\{\max\{x, 0\}, 1 - 1/|T|\}$ is a threshold function. Then we have the following Corollary:

**Corollary C.6.** *For a classification task T, the representations with both homophily and heterogeneity (e.g., $\hat{\mathbf{Z}}$) achieve a smaller supremum of Bayes error rate than the representations with only homophily (e.g., $\mathbf{Z}$) or with only heterogeneity (e.g., $\widetilde{\mathbf{Z}}$), i.e.,*

$$\sup(\bar{P}_e(\hat{\mathbf{Z}})) \leq \min(\sup(\bar{P}_e(\mathbf{Z})), \sup(\bar{P}_e(\widetilde{\mathbf{Z}}))). \tag{48}$$

*Proof.* According to the inequality between $\bar{P}_e(\hat{\mathbf{Z}})$ and $H(T|\hat{\mathbf{Z}})$ in previous work (Feder & Merhav, 1994), we first have

$$-\log(1 - \bar{P}_e(\hat{\mathbf{Z}})) \leq H(T \mid \hat{\mathbf{Z}}). \tag{49}$$

Based on Property 1 and Eq. (45), we have

$$\begin{aligned} H(T \mid \hat{\mathbf{Z}}) &= H(T) - I(T, \hat{\mathbf{Z}}) \\ &= H(T) - I(\mathbf{Z}, T) - I(\widetilde{\mathbf{Z}}, T|\mathbf{Z}). \end{aligned} \tag{50}$$

Similarly, we further have

$$H(T \mid \mathbf{Z}) = H(T) - I(T, \mathbf{Z}). \tag{51}$$

Based on Property 3, we have $H(T) - I(\mathbf{Z}, T) - I(\widetilde{\mathbf{Z}}, T|\mathbf{Z}) \leq H(T) - I(T, \mathbf{Z})$, and thus indicates that the R.H.S. of Eq. (50) is smaller than the R.H.S. of Eq. (51). Therefore, the fused representations $\hat{\mathbf{Z}}$ achieve a smaller supremum of Bayes error rate than $\mathbf{Z}$, *i.e.,* $\sup(\bar{P}_e(\hat{\mathbf{Z}})) \leq \sup(\bar{P}_e(\mathbf{Z}))$. Similarly, we can also obtain $\sup(\bar{P}_e(\hat{\mathbf{Z}})) \leq \sup(\bar{P}_e(\widetilde{\mathbf{Z}}))$. Therefore, we have $\sup(\bar{P}_e(\hat{\mathbf{Z}})) \leq \min(\sup(\bar{P}_e(\mathbf{Z})), \sup(\bar{P}_e(\widetilde{\mathbf{Z}})))$, and thus complete the proof. $\square$

## D EXPERIMENTAL SETTINGS

This section provides detailed experimental settings in Section 3, including the description of all datasets in Section D.1, summarization of all comparison methods in Section D.2, and model architectures and settings in Section D.3.

Table 3: Statistics of all datasets.

| Datasets | Type | #Nodes | #Node Types | #Edges | #Edge Types | Target Node | #Training | #Test |
|---|---|---|---|---|---|---|---|---|
| ACM | Heter | 8,994 | 3 | 25,922 | 4 | Paper | 600 | 2,125 |
| Yelp | Heter | 3,913 | 4 | 72,132 | 6 | Bussiness | 300 | 2,014 |
| DBLP | Heter | 18,405 | 3 | 67,946 | 4 | Author | 800 | 2,857 |
| Aminer | Heter | 55,783 | 3 | 153,676 | 4 | Paper | 80 | 1,000 |
| Freebase | Heter | 180,098 | 8 | 1,645,725 | 62 | Book | 1,909 | 5,568 |
| Amazon-Photo | Homo | 7,650 | 1 | 238,162 | 2 | Photo | 765 | 6,120 |
| Amazon-Computers | Homo | 13,752 | 1 | 491,722 | 2 | Computer | 1,375 | 11,002 |
| Coauthor-CS | Homo | 18,333 | 1 | 163,728 | 2 | Author | 1,833 | 14,667 |
| Coauthor-Physics | Homo | 34,493 | 1 | 495,924 | 2 | Author | 3,449 | 27,595 |

## D.1 DATASETS

We use five public heterogeneous graph datasets and four public homogeneous graph datasets from various domains. Heterogeneous graph datasets include three academic datasets (*i.e.,* ACM (Wang et al., 2019), DBLP (Wang et al., 2019), and Aminer (Hu et al., 2019)), one business dataset (*i.e.,* Yelp (Zhao et al., 2021a)), and one huge knowledge graph dataset (*i.e.,* Freebase (Lv et al., 2021)). Homogeneous graph datasets include two sale datasets (*i.e.,* Amazon-Photo and Amazon-Computers (Shchur et al., 2018)), and two co-authorship datasets (*i.e.,* Coauther-CS and Coauther-Physics (Sinha et al., 2015)). Table 3 summarizes the data statistics. We list the details of the datasets as follows.

- **ACM** is an academic heterogeneous graph dataset. It contains three types of nodes (paper (P), author (A), subject (S)), four types of edges (PA, AP, PS, SP), and categories of papers as labels.

- **Yelp** is a business heterogeneous graph dataset. It contains three types of nodes (business (B), user (U), service (S), level (L)), six types of edges (BU, UB, BS, SB, BL, LB), and categories of businesses as labels.

- **DBLP** is an academic heterogeneous graph dataset. It contains three types of nodes (paper (P), authors (A), conference (C)), four types of edges (PA, AP, PC, CP), and research areas of authors as labels.

- **Aminer** is an academic heterogeneous graph dataset. It contains three types of nodes (paper (P), author (A), reference (R)), four types of edges (PA, AP, PR, RP), and categories of papers as labels.

- **Freebase** is a huge knowledge heterogeneous graph dataset. It contains eight types of nodes (book (B), film (F), location (L), music (M), person (P), sport (S), organization (O), business (U)), 62 types of edges, and categories of books as labels.

- **Amazon-Photo** and **Amazon-Computers** are two co-purchase homogeneous graph datasets. They are two networks extracted from Amazon's co-purchase data. Nodes are products, and edges denote that these products were often bought together. Products are categorized into several classes by the product category.

- **Coauthor-CS** and **Coauthor-Physics** are two co-author homogeneous graph datasets. They are two networks extracted from the Microsoft Academic Graph (22). Nodes are authors, and edges denote a collaboration of two authors. Authors are categorized into several classes by research fields.

## D.2 COMPARISON METHODS

The comparison methods include eleven heterogeneous graph methods and twelve homogeneous graph methods. Heterogeneous graph methods include Mp2vec (Dong et al., 2017), HAN (Wang et al., 2019), HGT (Hu et al., 2020), DMGI (Park et al., 2020), DMGIattn (Park et al., 2020), HDMI

Table 4: The characteristics of all comparison methods.

| Methods | Heterogeneous | Homogeneous | Semi-sup | Self-sup/unsup | Meta-path |
|---|---|---|---|---|---|
| DeepWalk (2014) | | ✓ | | ✓ | |
| GCN (2017) | | ✓ | ✓ | | |
| GAT (2018) | | ✓ | ✓ | | |
| DGI (2019) | | ✓ | | ✓ | |
| GMI (2020) | | ✓ | | ✓ | |
| MVGRL (2020) | | ✓ | | ✓ | |
| GRACE (2020) | | ✓ | | ✓ | |
| GCA (2021) | | ✓ | | ✓ | |
| GIC (2021) | | ✓ | | ✓ | |
| G-BT (2022) | | ✓ | | ✓ | |
| COSTA (2022) | | ✓ | | ✓ | |
| DSSL (2022) | | ✓ | | ✓ | |
| Mp2vec (2017) | ✓ | | | ✓ | ✓ |
| HAN (2019) | ✓ | | ✓ | | ✓ |
| HGT (2020) | ✓ | | ✓ | | |
| DMGI (2020) | ✓ | | | ✓ | ✓ |
| DMGIattn (2020) | ✓ | | | ✓ | ✓ |
| HDMI (2021) | ✓ | | | ✓ | ✓ |
| HeCo (2021) | ✓ | | | ✓ | ✓ |
| HGCML (2023) | ✓ | | | ✓ | ✓ |
| CPIM (2023) | ✓ | | | ✓ | ✓ |
| HGMAE (2023) | ✓ | | | ✓ | ✓ |
| DMG (2023) | ✓ | | | ✓ | ✓ |
| HERO (ours) | ✓ | | | ✓ | |

(Jing et al., 2021), HeCo (Wang et al., 2021), HGCML (Wang et al., 2023), CPIM (Mo et al., 2023b), HGMAE (Tian et al., 2023), and DMG (Mo et al., 2023a). Homogeneous graph methods include GCN (Kipf & Welling, 2017), GAT (Velickovic et al., 2018), DeepWalk (Perozzi et al., 2014), DGI (Velickovic et al., 2019), GMI (Peng et al., 2020), MVGRL (Hassani & Khasahmadi, 2020), GRACE (Zhu et al., 2020b), GCA (Zhu et al., 2021), GIC (Mavromatis & Karypis, 2021), G-BT (Bielak et al., 2022), COSTA (Zhang et al., 2022b), and DSSL (Xiao et al., 2022). The characteristics of all methods are listed in Table 4, where "Heterogeneous" and "Homogeneous" indicate the methods designed for the heterogeneous graph and homogeneous graph, respectively. "Semi-sup", and "Self-sup/unsup" indicate that the method conducts semi-supervised learning, and self-supervised/unsupervised learning, respectively. "Meta-path" indicates that the method requires pre-defined meta-paths during the training process.

Table 5: Similarity search performance (*i.e.,* Sim@5 and Sim@10) on heterogeneous graph datasets.

| Method | ACM | | Yelp | | DBLP | | Aminer | |
|---|---|---|---|---|---|---|---|---|
| | Sim@5 | Sim@10 | Sim@5 | Sim@10 | Sim@5 | Sim@10 | Sim@5 | Sim@10 |
| Deep Walk | 78.7±0.2 | 76.7±0.3 | 73.6±0.4 | 72.1±0.7 | 85.7±0.2 | 84.6±0.3 | 67.1±0.4 | 65.9±0.6 |
| GCN | 86.8±0.3 | 84.9±0.3 | 85.1±0.5 | 83.9±0.4 | 88.2±0.1 | 87.4±0.2 | 77.4±0.4 | 75.2±0.5 |
| GAT | 86.5±0.2 | 85.4±0.4 | 85.3±0.3 | 84.2±0.5 | 90.6±0.3 | 90.2±0.3 | 76.8±0.5 | 74.6±0.6 |
| Mp2vec | 82.3±0.4 | 81.2±0.5 | 79.1±0.6 | 77.9±0.7 | 87.1±0.5 | 85.8±0.6 | 71.8±0.3 | 68.6±0.4 |
| HAN | 87.2±0.2 | 85.6±0.3 | 87.5±0.7 | 87.4±0.4 | 89.3±0.3 | 89.3±0.5 | 78.2±0.4 | 78.0±0.5 |
| HGT | **90.1±0.5** | 90.0±0.6 | 88.6±0.7 | 88.0±0.5 | 90.3±0.4 | 89.9±0.7 | 79.2±0.7 | 78.3±0.6 |
| DMGI | 89.8±0.3 | 89.1±0.1 | 83.2±0.5 | 82.5±0.7 | 91.3±0.4 | 91.0±0.3 | 76.3±0.8 | 73.3±0.7 |
| DMGIattn | 90.1±0.2 | 88.9±0.4 | 80.7±0.4 | 79.6±0.2 | 89.0±0.2 | 87.9±0.4 | 72.8±0.6 | 71.5±0.5 |
| HDMI | 89.8±0.4 | 89.0±0.2 | 83.2±0.4 | 79.8±0.7 | 91.2±0.3 | 90.9±0.2 | 78.5±0.5 | 77.1±0.7 |
| HeCo | 89.6±0.2 | 88.7±0.2 | 86.4±0.3 | 85.1±0.4 | 90.9±0.2 | 90.6±0.1 | 82.9±0.7 | 81.7±0.5 |
| HGCML | 89.5±0.5 | 89.0±0.6 | 87.1±0.6 | 85.8±0.7 | 91.3±0.4 | 90.8±0.6 | 81.7±0.6 | 80.5±0.4 |
| CPIM | 89.6±0.6 | 88.8±0.4 | 87.9±0.7 | 86.8±0.5 | 91.1±0.6 | 91.0±0.3 | 82.1±0.6 | 80.9±0.8 |
| HGMAE | 89.5±0.3 | 88.9±0.5 | 88.1±0.8 | 86.9±0.7 | 90.9±0.4 | 90.7±0.5 | 80.5±0.4 | 79.5±0.6 |
| DMG | 89.2±0.6 | 88.4±0.5 | 88.3±0.6 | 87.5±0.3 | 91.4±0.5 | 91.0±0.7 | 83.1±0.4 | 81.8±0.5 |
| **HERO** | 89.7±0.6 | **89.2±0.4** | **89.7±0.3** | **88.9±0.5** | **92.1±0.7** | **91.9±0.5** | **86.8±0.3** | **85.1±0.6** |

Table 6: Classification performance (*i.e.,* Macro-F1 and Micro-F1) of different components of the objective function on heterogeneous graph datasets.

| Method | ACM | | Yelp | | DBLP | | Aminer | |
|---|---|---|---|---|---|---|---|---|
| | Macro-F1 | Micro-F1 | Macro-F1 | Micro-F1 | Macro-F1 | Micro-F1 | Macro-F1 | Micro-F1 |
| w/o $\mathcal{L}_{con}$ | 89.9±0.5 | 89.8±0.7 | 91.2±0.5 | 91.0±0.3 | 88.0±0.6 | 89.1±0.5 | 74.6±0.7 | 84.0±0.9 |
| w/o $\mathcal{L}_{spe}$ | 90.4±0.3 | 90.3±0.5 | 91.2±0.4 | 88.5±0.7 | 92.1±0.6 | 93.1±0.5 | 74.1±0.7 | 83.1±0.5 |
| **HERO** | **92.2±0.5** | **92.1±0.7** | **92.4±0.7** | **92.3±0.6** | **93.8±0.6** | **94.4±0.4** | **75.1±0.7** | **84.5±0.9** |

### D.3 MODEL ARCHITECTURES AND SETTINGS

As described in Section 2, the proposed method employs the MLP (*i.e.,* $g_\phi$) and the closed-form solution of the self-expressive matrix $\mathbf{S}^*$ to obtain the homophilous representations $\mathbf{Z}$. Moreover, the proposed method employs the heterogeneous encoder (*i.e.,* $f_\theta$) to obtain heterogeneous representations $\widetilde{\mathbf{Z}}$. After that, the proposed method employs projection head $p_\varphi$ and $q_\gamma$ to map the homophilous representations and heterogeneous representations into latent spaces. In the proposed method, projection head $p_\varphi$ and $q_\gamma$ are simply implemented by the one linear layer, followed by a ReLU activation. In addition, the proposed method employs a transformation $u_\phi$ to avoid directly aligning the homophilous and heterogeneous representations. The transformation $u_\phi$ is also implemented by the one linear layer, followed by a ReLU activation. Finally, In the proposed method, all parameters were optimized by the Adam optimizer (Kingma & Ba, 2015) with an initial learning rate. Moreover, We use early stopping with a patience of 30 to train the proposed SHGL model. In all experiments, we repeat the experiments five times for all methods and report the average results.

### D.4 EVALUATION PROTOCOL

We follow the evaluation in previous works (Jing et al., 2021; Pan & Kang, 2021; Zhou et al., 2022) to conduct node classification and similarity search as semi-supervised and unsupervised downstream tasks, respectively. Specifically, we first pre-train models with unlabeled data in a self-supervised manner and output learned node representations. After that, the resulting representations can be used for different downstream tasks. For the node classification task, we train a simple logistic regression classifier with a fixed iteration number, and evaluate the effectiveness of all methods with Micro-F1 and Macro-F1 scores. For the similarity search task, we first compute the cosine similarity between the representations of all node pairs, then calculate the ratio of possessing the same label within the top-5 and top-10 most similar node pairs (Sim@5 and Sim@10).

## E ADDITIONAL EXPERIMENTS

This section provides some additional experimental results to support the proposed method (*i.e.,* HERO), including experiments on the large-scale dataset in Section E.1, ablation studies in Section E.2-E.8, visualization of self-expressive matrix in Section E.9, parameter analysis in Section E.10, comparison experiments on the similarity search task in Table 5.

Table 7: Classification performance (*i.e.,* Macro-F1 and Micro-F1) of different representations on heterogeneous graph datasets.

| Method | ACM | | Yelp | | DBLP | | Aminer | |
|---|---|---|---|---|---|---|---|---|
| | Macro-F1 | Micro-F1 | Macro-F1 | Micro-F1 | Macro-F1 | Micro-F1 | Macro-F1 | Micro-F1 |
| w/o $\mathbf{Z}$ | 91.2±0.7 | 91.1±0.5 | 91.1±0.5 | 91.3±0.6 | 93.1±0.7 | 93.7±0.6 | 72.4±0.5 | 81.9±0.8 |
| w/o $\widetilde{\mathbf{Z}}$ | 87.8±0.6 | 87.8±0.4 | 89.3±0.4 | 89.4±0.5 | 74.6±0.5 | 75.5±0.7 | 58.5±0.8 | 63.7±0.6 |
| **HERO** | **92.2±0.5** | **92.1±0.7** | **92.4±0.7** | **92.3±0.6** | **93.8±0.6** | **94.4±0.4** | **75.1±0.7** | **84.5±0.9** |

Table 8: Classification performance (*i.e.,* Macro-F1 and Micro-F1) of the subspace and nearby neighbors homophily objective function on heterogeneous graph datasets.

| Method | ACM | | Yelp | | DBLP | | Aminer | |
|---|---|---|---|---|---|---|---|---|
| | Macro-F1 | Micro-F1 | Macro-F1 | Micro-F1 | Macro-F1 | Micro-F1 | Macro-F1 | Micro-F1 |
| w/o $\mathcal{L}_{sub}$ | 86.8±0.4 | 86.6±0.6 | 91.3±0.7 | 90.8±0.5 | 92.5±0.6 | 93.4±0.8 | 69.8±0.5 | 76.8±0.6 |
| w/o $\mathcal{L}_{nei}$ | 91.4±0.5 | 91.3±0.3 | 91.8±0.7 | 91.3±0.6 | 93.0±0.4 | 94.0±0.5 | 71.4±0.6 | 79.8±0.3 |
| **HERO** | **92.2±0.5** | **92.1±0.7** | **92.4±0.7** | **92.3±0.6** | **93.8±0.6** | **94.4±0.4** | **75.1±0.7** | **84.5±0.9** |

Table 9: Classification performance (*i.e.,* Macro-F1 and Micro-F1) of self-expressive and self-attention mechanisms on heterogeneous graph datasets.

| Method | ACM | | Yelp | | DBLP | | Aminer | |
|---|---|---|---|---|---|---|---|---|
| | Macro-F1 | Micro-F1 | Macro-F1 | Micro-F1 | Macro-F1 | Micro-F1 | Macro-F1 | Micro-F1 |
| self-attention | 88.7±0.8 | 88.4±0.7 | 92.0±0.5 | 91.7±0.6 | 91.2±0.4 | 92.1±0.6 | 73.2±0.7 | 82.1±0.6 |
| self-expressive | **92.2±0.5** | **92.1±0.7** | **92.4±0.7** | **92.3±0.6** | **93.8±0.6** | **94.4±0.4** | **75.1±0.7** | **84.5±0.9** |

### E.1 EFFECTIVENESS AND EFFICIENCY ON THE LARGE-SCALE DATASET

The time complexity of the proposed method is linearly related to the sample size, which shows its potential to be applied to large-scale datasets. To further verify the effectiveness and efficiency of the proposed method on the large-scale dataset, we evaluate the node classification performance, training time, and memory cost of the proposed method and all SHGL comparison methods on the huge knowledge graph dataset (*i.e.,* Freebase) and report the results in Figure 5. From Figure 5, we have the observations as follows. First, the proposed method achieves the best classification performance and the minimal training time cost, compared to other SHGL comparison methods. This can be attributed to the fact that the time complexity of the proposed method is linearly with the sample size while other comparison methods generally require quadratic time complexity. Second, the proposed method also achieves the minimal memory cost on the huge knowledge graph dataset. This verifies the scalability of the proposed method on the large-scale dataset.

### E.2 EFFECTIVENESS OF DIFFERENT COMPONENTS

The proposed method investigates the consistency loss (*i.e.,* $\mathcal{L}_{con}$), and the specificity loss (*i.e.,* $\mathcal{L}_{spe}$) to extract the consistent information between homophilous and heterogeneous representations and preserve their specific information, respectively. To verify the effectiveness of each component of the objective function in the proposed method, we investigate the performance of all variants of the objective function on the node classification task and report the results in Table 6. Moreover, we also investigate the performance of using only homophilous representations or heterogeneous representations on the node classification task and report the results in Table 7.

From Tables 6 and 7, we have the observations as follows. First, the proposed method with the complete objective function achieves the best performance. For example, the proposed method on average improves by 2.7% and 2.0%, compared to the variant method without $\mathcal{L}_{con}$ and the variant method without $\mathcal{L}_{spe}$, respectively, indicating that all components of the objective function are both significant for the proposed method. This suggests that both the consistent information and specific information are important for downstream tasks. Second, the variant method that only applies homophilous representations or heterogeneous representations to downstream tasks obtains inferior performance, compared to the proposed method that both uses homophilous representations and heterogeneous representations. This makes sense as the homophilous representations and heterogeneous representations can complement each other, thus benefiting downstream tasks.

### E.3 EFFECTIVENESS OF THE SELF-EXPRESSIVE MECHANISM

The proposed method employs the self-expressive mechanism to capture the homophily from the subspace and nearby neighbors. Actually, the self-expressive mechanism shares a similar idea with

Table 10: Classification performance (*i.e.,* Macro-F1 and Micro-F1) of the same and different projection heads on heterogeneous graph datasets.

| Method | ACM | | Yelp | | DBLP | | Aminer | |
|---|---|---|---|---|---|---|---|---|
| | Macro-F1 | Micro-F1 | Macro-F1 | Micro-F1 | Macro-F1 | Micro-F1 | Macro-F1 | Micro-F1 |
| different pro | 89.6±0.3 | 89.4±0.7 | 92.2±0.4 | 92.0±0.5 | 92.9±0.7 | 93.9±0.5 | 71.0±0.4 | 79.4±0.5 |
| same pro | **92.2±0.5** | **92.1±0.7** | **92.4±0.7** | **92.3±0.6** | **93.8±0.6** | **94.4±0.4** | **75.1±0.7** | **84.5±0.9** |

Table 11: Classification performance (*i.e.,* Macro-F1 and Micro-F1) of proposed method with the GCN encoder and GAT encoder on homogeneous graph datasets.

| Method | Amazon-Photo | | Amazon-Computer | | Coauthor-CS | | Coauthor-Physics | |
|---|---|---|---|---|---|---|---|---|
| | Macro-F1 | Micro-F1 | Macro-F1 | Micro-F1 | Macro-F1 | Micro-F1 | Macro-F1 | Micro-F1 |
| GAT | 91.2±0.5 | 92.6±0.7 | **86.1±0.3** | 88.2±0.4 | **90.8±0.6** | 92.9±0.4 | 94.4±0.5 | 95.8±0.4 |
| GCN | **91.8±0.4** | **93.0±0.3** | 85.7±0.6 | **88.4±0.5** | 90.6±0.5 | **93.3±0.6** | **94.6±0.7** | **96.0±0.5** |

the self-attention mechanism, *i.e.,* linearly describing each sample with all samples. We analyzed the major difference between them in Section A.3. To further verify the effectiveness of the self-expressive mechanism, we investigate the performance of the variants methods with the self-expressive and self-attention mechanisms and report the results in Table 9. Obviously, the self-expressive mechanism obtains better performance than the self-attention mechanism on all datasets. The reason can be summarized as follows. First, the self-attention mechanism is restricted to be non-negative, while the self-expressive mechanism is not restricted. Therefore, the self-expressive matrix is able to capture the positive relationships among nodes within the same class and the negative relationships among nodes from different classes, respectively. Second, the self-expressive matrix in the proposed method is encouraged to focus on the most relevant neighbors of each sample instead of arbitrary nodes in the self-attention mechanism. Therefore, the self-expressive matrix focuses more on the neighbors of each node that may come from the same class and less on its faraway nodes that may come from different classes. As a result, the effectiveness of the self-expressive mechanism is verified.

### E.4 EFFECTIVENESS OF DIFFERENT DIMENSIONS

According to the complexity analysis in Section B.2, the time complexity of the proposed method is $\mathcal{O}(Nd^2 + d^3 + kd)$, which is highly related to the representation dimension $d$. Therefore, we conduct an ablation study on the proposed method with different dimensions on four heterogeneous graph datasets and report the results in Figure 6. From Figure 6, we have the observations as follows. First, the proposed method generally achieves the best performance with the dimension set to 128 or 256. Second, the performance of the proposed method is stable as the dimension increases. This indicates that the proposed method does not require large dimensions to achieve significant performance. Generally, the representation dimensions satisfy that $d^2 \leq N$, where $N$ is the number of nodes. Therefore, this further verifies the efficiency of the proposed method.

### E.5 EFFECTIVENESS OF THE SUBSPACE AND NEARBY NEIGHBOR HOMOPHILY

The proposed method investigates the first term in Eq. (3) to optimize the self-expressive matrix $\mathbf{S}$ to capture the homophily in the same subspace while investigates the second term to enable the self-expressive matrix $\mathbf{S}$ to capture the homophily among the node and its nearby neighbors. To verify the effectiveness of two terms in Eq. (3), we denote the first term and the second term as $\mathcal{L}_{sub}$ and $\mathcal{L}_{nei}$, respectively. Then we investigate the performance of variants methods with $\mathcal{L}_{sub}$ or $\mathcal{L}_{nei}$ only and report the classification results in Table 8.

First, the proposed method considering both the homophily from the subspace and nearby neighbor obtains the best performance. For example, the proposed method on average improves by 2.2% and 4.6%, compared to the variant method that only considers the homophily in the subspace or nearby neighbors. This indicates that both the homophily in the subspace and nearby neighbors are essential for the proposed method and can complement each other, enforcing that the nodes within

Table 12: Classification performance (*i.e.,* Macro-F1 and Micro-F1) different fusion mechanisms on heterogeneous graph datasets.

| Method | ACM | | Yelp | | DBLP | | Aminer | |
|---|---|---|---|---|---|---|---|---|
| | Macro-F1 | Micro-F1 | Macro-F1 | Micro-F1 | Macro-F1 | Micro-F1 | Macro-F1 | Micro-F1 |
| Average Pooling | **92.3±0.2** | **92.1±0.4** | 92.2±0.6 | 92.0±0.5 | 93.3±0.3 | 94.1±0.5 | 74.9±0.7 | 83.2±0.6 |
| Max Pooling | 91.2±0.5 | 91.0±0.7 | 90.9±0.3 | 90.6±0.4 | 91.9±0.7 | 92.2±0.8 | 71.1±0.5 | 80.5±0.4 |
| Mix Pooling | 92.0±0.6 | 91.9±0.5 | 91.8±0.5 | 91.9±0.7 | 92.8±0.6 | 93.9±0.4 | 74.5±0.7 | 83.7±0.8 |
| Concatenation | 92.2±0.5 | **92.1±0.7** | **92.4±0.7** | **92.3±0.6** | **93.8±0.6** | **94.4±0.4** | **75.1±0.7** | **84.5±0.9** |

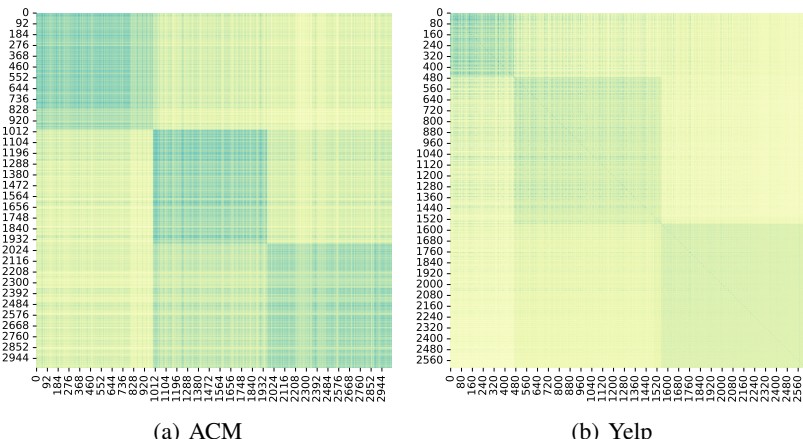

(a) ACM          (b) Yelp

Figure 4: (a) and (b) indicate the self-expressive matrix heatmaps of ACM and Yelp datasets reordered by node labels.

the same class have similar representations and benefit downstream tasks. Second, the variant method with $\mathcal{L}_{sub}$ only obtains better performance compared to the variant method with $\mathcal{L}_{nei}$ only. This demonstrates that $\mathcal{L}_{sub}$ in Eq. (3) captures main homophily in the subspace, and $\mathcal{L}_{nei}$ in Eq. (3) provides some complementary homophily from the nearby neighbors.

### E.6 EFFECTIVENESS OF THE PROJECTION HEAD

The proposed method uses the same projection head to map homophilous and heterogeneous representations into the same latent space. To verify the effectiveness of the projection head, we investigate the performance of variant methods with the same projection head and different projection head and report the results in Table 10. Obviously, the variant method with the same projection head obtains the best performance. The reason can be attributed to the fact that the same projection head can map homophilous and heterogeneous representations into the same space, so that they are comparable in the same latent space.

### E.7 EFFECTIVENESS OF THE ENCODER ON HOMOGENEOUS GRAPH

The proposed method replaces the heterogeneous encoder with GCN on homogeneous graph datasets, because the heterogeneous encoder is used to deal with multiple node types in the heterogeneous graph, while there is one node type in the homogeneous graph. To study the impact of the encoders in the proposed method, we conduct experiments for variant methods using different encoders (*i.e.,* GCN and GAT) and report the results in Table 11. The results indicate that the variant methods with GCN encoder and GAT encoder show similar performance on all datasets, demonstrating that the proposed method is robust to different encoders.

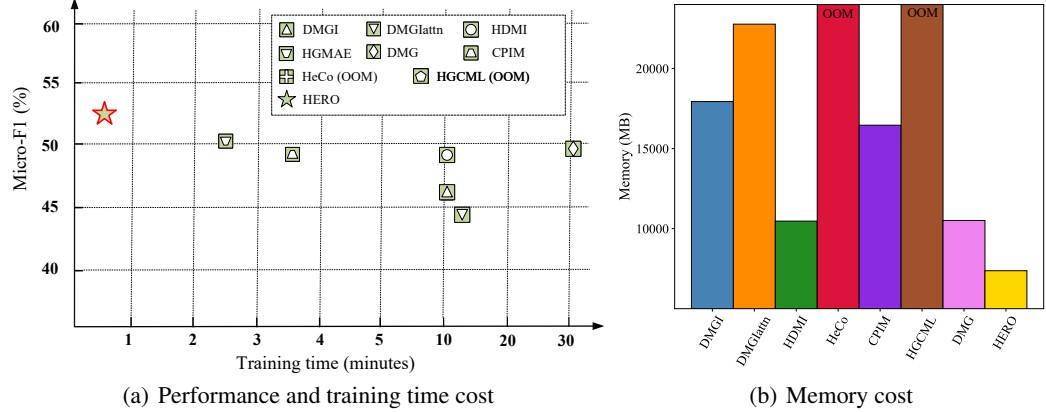

(a) Performance and training time cost

(b) Memory cost

Figure 5: The classification performance, training time cost, and memory cost of the proposed HERO and all SHGL comparison methods on the huge knowledge graph dataset Freebase, where OOM indicates Out-Of-Memory.

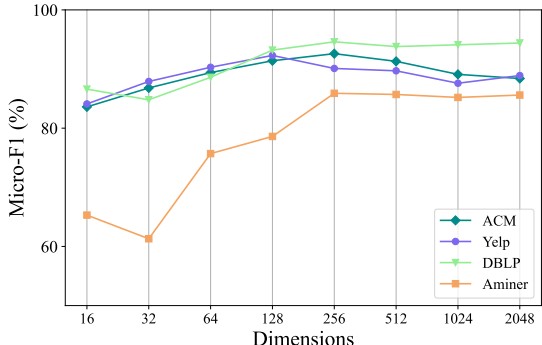

Figure 6: The classification performance of the proposed method with different representation dimensions on four heterogeneous graph datasets.

### E.8 EFFECTIVENESS OF THE FUSION MECHANISM

The proposed method employs a simple concatenation mechanism to fuse homophilous representations and heterogeneous representations for downstream tasks. To study the impact of the fusion mechanism, we investigate the performance of the proposed method with different fusion mechanisms (*i.e.,* average pooling (LeCun et al., 1989), max pooling (Murray & Perronnin, 2014), mix pooling (Yu et al., 2014), and concatenation) and report the results in Table 12. Obviously, the concatenation mechanism obtains the superior results. This indicates that the proposed method requires only a very simple fusion mechanism (*e.g.,* concatenation) to achieve good performance, without the need for complex fusion operations.

### E.9 VISUALIZATION OF THE SELF-EXPRESSIVE MATRIX

The proposed method employs the self-expressive matrix to capture the homophily from the subspace and nearby neighbors. To verify that the self-expressive matrix indeed captures the homophily in the heterogeneous graph, we visualize the self-expressive matrix heatmaps of ACM and Yelp datasets in Figure 4, where rows and columns are reordered by node labels. In the heatmaps, the darker a pixel, the larger the value of self-expressive matrix weight. From Figure 4, we observe that the heatmaps exhibit a similar block diagonal structure with the correlation map of homophilous representations. This indicates that the self-expressive matrix assigns large weights for nodes from the same class and

small weights for nodes from different classes to describe each node. Therefore, the self-expressive indeed captures the homophily in the heterogeneous graph.

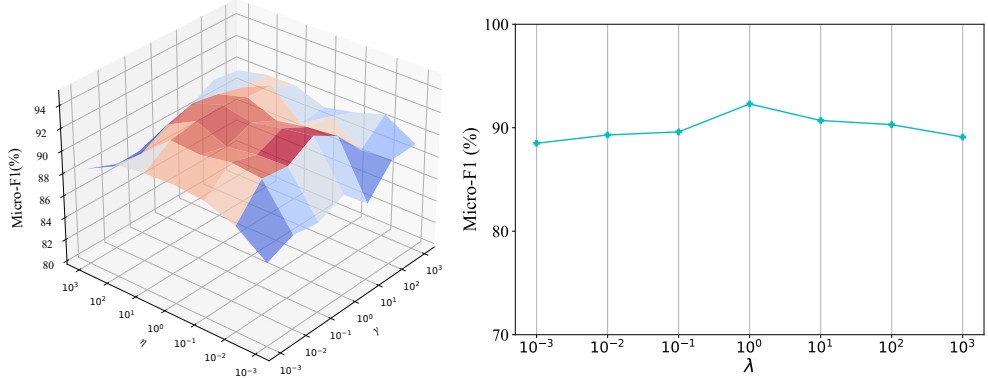

Figure 7: The classification performance of the proposed method at different parameter settings (*i.e.,* $\gamma$, $\eta$, and $\lambda$) on the ACM dataset.

### E.10 PARAMETER ANALYSIS

In the proposed method, we employ the non-negative parameters (*i.e.,* $\gamma$, $\eta$, and $\lambda$) to achieve a trade-off between each term of the consistency loss, specificity loss, and the final objective function. To investigate the impact of $\gamma$, $\eta$, and $\lambda$ with different settings, we conduct the node classification on the ACM datasets by varying the value of parameters in the range of $[10^{-3},10^3]$ and reporting the results in Figure 7.

From Figure 7, the observations can be summarized as follows. First, for the parameters $\gamma$ and $\eta$, the proposed method consistently achieves significant performance within the range of $[10^{-2},10^2]$. Moreover, if the values of parameters are too large (*e.g.,* $> 10^2$) or too small (*e.g.,* $< 10^{-2}$), the proposed method obtains inferior performance. This indicates that each term in the consistency loss and specificity loss is essential for the proposed method. Second, for the parameter $\lambda$, the proposed method achieves the best results while the value of the parameter is set in the range of $[10^{-1},10^1]$. This further confirms the importance of both consistency and specificity loss for the proposed method.

## F POTENTIAL LIMITATIONS AND FUTURE DIRECTIONS

In this paper, we employ the closed-form solution of the self-expressive matrix to describe and capture the homophily in the heterogeneous graph from the subspace and nearby neighbors, thus benefiting downstream tasks. Actually, it is worth noting that various methods can be utilized to capture the interconnections between nodes within the same class, thereby effectively describing the homophily within the heterogeneous graph. Furthermore, the proposed method relies on node features to calculate the feature distance between all pairs of nodes. However, there are cases in the heterogeneous graph where nodes lack features. Although one-hot vectors or structural embedding can be assigned as features to address this issue, we acknowledge the need to develop specific methods tailored for the heterogeneous graph without node features to further enhance effectiveness. We consider these aspects as potential directions for future research.

