# OpenReview forum: "Self-Supervised Heterogeneous Graph Learning:  a Homophily and Heterogeneity View"
_ICLR.cc/2024/Conference — ICLR 2024 poster_

### Official Review · Reviewer_BScY · 2023-10-24

**Soundness:** 3 good
**Presentation:** 3 good
**Contribution:** 3 good
**Rating:** 6
**Confidence:** 4

**Summary:**

To capture homogeneity and heterogeneity in the heterogeneous graph without pre-defined meta-paths, this work proposes to adaptively learn a self-expressive matrix and employ the heterogeneous encoder to obtain homogeneous and heterogeneous representations. In addition, the proposed method designs the objective function to extract the consistent information between homogeneous representations and heterogeneous representations and to maintain their specific information in different latent spaces. Moreover, the authors theoretically demonstrate the effectiveness of the proposed method and support the claims made in this paper.

**Strengths:**

a)Idea in this work is overall novel and attractive, and it will inspire researchers in related fields to explore new methods to capture the homogeneity in the heterogeneous graph without meta-paths, which induces inconveniences and large computation costs.

b)Extensive experimental results demonstrate the effectiveness of the proposed method on both homogeneous and heterogeneous graph datasets in terms of different downstream tasks.

c)Claims in this paper are supported well. That is, the learned homogeneous representations are demonstrated to exhibit the grouping effect to capture the homogeneity, and considering both homogeneity and heterogeneity introduces more task-related information to benefit downstream tasks.

**Weaknesses:**

a)Not clear why the concatenation mechanism is directly employed to fuse homogeneous representation and heterogeneous representations to obtain final representations for downstream tasks. What would be the impact on the performance of the method if other fusion methods were used, such as average pooling? The authors could add such ablation studies.

b)It would be better to add some details of experiments. For example, the detailed information of all datasets used in this paper.

**Questions:**

a)The effectiveness of the proposed HERO and other comparison methods are evaluated through two downstream tasks, namely, node classification and similarity search. What are the significant distinctions between these tasks?

b)Theorem 2.3 does not constrain the type of downstream tasks. Does it mean that the proposed method is expected to achieve better performance on different downstream tasks, compared to the methods considering the homogeneity only?

Overall, this paper holds considerable value and potential, and I will increase my rating if the weaknesses and questions can be addressed and discussed.

---

> ### Author Response · Authors · 2023-11-18
> **Response to Reviewer BScY**
>
> We thank the reviewer for the insightful comments!
>
> > **W1**. Not clear why the concatenation mechanism is directly employed to fuse homogeneous representation and heterogeneous representations to obtain final representations for downstream tasks. What would be the impact on the performance of the method if other fusion methods were used, such as average pooling? The authors could add such ablation studies.
>
> Thanks for your constructive suggestions. In the revision, to study the impact of the fusion mechanism, we investigated the performance of the proposed method with different fusion mechanisms (i.e., average pooling, max pooling, mix pooling, and concatenation) and reported the results in Table 12 in Appendix E.8. Obviously, the concatenation mechanism obtains the superior results. This indicates that the proposed method requires only a very simple fusion mechanism (e.g., concatenation) to achieve good performance without the need for complex fusion operations.
>
> > **W2**. It would be better to add some details of experiments. For example, the detailed information of all datasets used in this paper.
>
> In the manuscript, the detailed information of all datasets is listed in Table 3 in Appendix D.1.
>
> > **Q1**. The effectiveness of the proposed HERO and other comparison methods are evaluated through two downstream tasks, namely, node classification and similarity search. What are the significant distinctions between these tasks?
>
> There are major differences between them. The node classification is a semi-supervised task, while the similarity search is an unsupervised task. Specifically, node classification focuses on whether each node is correctly categorized on its own, while similarity search focuses on the ratio of possessing the same label within the most similar node pairs.
>
> > **Q2**. Theorem 2.3 does not constrain the type of downstream tasks. Does it mean that the proposed method is expected to achieve better performance on different downstream tasks, compared to the methods considering the homogeneity only?
>
> Yes, we don’t constrain the type of downstream tasks in Theorem 2.3. That is, the proposed method is expected to introduce more task-related information for different downstream tasks, and thus benefit their performance, compared to previous SHGL methods. This is further verified in the Experiment section of the manuscript.

---

### Official Review · Reviewer_rAzi · 2023-10-26

**Soundness:** 3 good
**Presentation:** 3 good
**Contribution:** 4 excellent
**Rating:** 8
**Confidence:** 4

**Summary:**

* This work captures both homogeneity and heterogeneity in the heterogeneous graph without pre-defined meta-paths.

* A self-expressive matrix is used to capture the homogeneity from the subspace and nearby neighbors. Meanwhile, the heterogeneity is captured by aggregating the information of nodes from different types.

* The proposed method extracts the consistent information between homogeneity and heterogeneity and preserves their specific task-related information, leading to the effectiveness.

* The experimental results show that the proposed method achieves SOTA in various datasets, compared to numerous comparison methods.

**Strengths:**

* The motivation of the proposed method is clearly stated with the illustrative examples and empirical studies. I really appreciate the examples used in the Introduction. Moreover, the design of the framework is easy to understand.

* The method part is solid, and exploring homogeneity and heterogeneity in the heterogeneous graph without predefined meta-paths makes sense.

* Theoretical analysis verifies that the proposed method captures the homogeneity in the heterogeneous graph and is expected to obtain better performance than previous methods.

**Weaknesses:**

* A corollary is shown in the Appendix to verify that the representations with both homogeneity and heterogeneity indeed obtain a better downstream task performance than the representations with homogeneity or heterogeneity only. I think this is a strong addition to Theorem 2.3 and should be mentioned in the main text.

* The proposed method seems to yield overall smaller improvements on homogeneous graph datasets than on heterogeneous graph datasets when compared to comparative methods.

**Questions:**

* The idea of capturing the homogeneity from the subspace and nearby neighbors is interesting. Can it be used in other related domains, such as computer vision?

* Why replace the heterogeneous encoder with GCN to implement the proposed method on homogeneous graph datasets? What happens if the heterogeneous encoder is replaced with other similar encoders, e.g., GAT?

---

> ### Author Response · Authors · 2023-11-18
> **Response to Reviewer rAzi**
>
> We thank the reviewer for the insightful comments!
>
> > **W1**. A corollary is shown in the Appendix to verify that the representations with both homogeneity and heterogeneity indeed obtain a better downstream task performance than the representations with homogeneity or heterogeneity only. I think this is a strong addition to Theorem 2.3 and should be mentioned in the main text.
>
> Thanks for your suggestion. In the revision, we mentioned the Corollary in the main text.
>
> > **W2**. The proposed method seems to yield overall smaller improvements on homogeneous graph datasets than on heterogeneous graph datasets when compared to comparative methods.
>
> The reason can be attributed to the fact that the proposed method is designed for the heterogeneous graph. Specifically, the proposed method maintains the task-related information of the heterogeneity and homogeneity for the heterogeneous graph, while there is no heterogeneity in the homogeneous graph. However, even for the homogeneous graph datasets, the proposed method still obtains competitive performance, compared to the comparison methods.
>
> > **Q1**. The idea of capturing the homogeneity from the subspace and nearby neighbors is interesting. Can it be used in other related domains, such as computer vision?
>
> Yes, it can be easily used in other domains. Take the image recognition as an example, the homogeneity from the subspace and nearby neighbors can be used to capture the relationships among different patches of an image.
>
> > **Q2**. Why replace the heterogeneous encoder with GCN to implement the proposed method on homogeneous graph datasets? What happens if the heterogeneous encoder is replaced with other similar encoders, e.g., GAT?
>
> Because the heterogeneous encoder is used to deal with multiple node types in the heterogeneous graph, while there is only one node type in the homogeneous graph. Therefore, we follow previous self-supervised works and select GCN as the encoder to obtain node representations for the homogeneous graph. In the revision, to study the impact of encoders in the proposed method, we investigated the performance of variant methods using different encoders (i.e., GCN and GAT) and reported the results in Table 11 in Appendix E.7. The results indicate that the variant methods with GCN encoder and GAT encoder show similar performance on all datasets, demonstrating that the proposed method is robust to different encoders on the homogeneous graph.

---

### Official Review · Reviewer_EpeM · 2023-10-31

**Soundness:** 4 excellent
**Presentation:** 3 good
**Contribution:** 3 good
**Rating:** 8
**Confidence:** 5

**Summary:**

This paper proposes to understand the self-supervised heterogeneous graph learning without pre-defined meta-paths from the perspective of homogeneity and heterogeneity. To do this, this paper captures the homogeneity from both the subspace and nearby neighbors as well as to discard pre-defined meta-paths. Moreover, the proposed method further extracts the consistent and specific contents between homogeneous and heterogeneous representations to introduce more task-related information. Experimental results and theoretical analysis demonstrate the superiority of the proposed method.

**Strengths:**

1.	The authors provide a deeper insight (i.e., heterogeneity mining and homogeneity mining) into existing self-supervised heterogeneous graph representation learning, which is interesting and reasonable.

2.	This paper proposes to use the self-expression matrix rather than traditional meta-paths to capture homogeneity in the heterogeneous graph, which opens up a new possibility for the self-supervised heterogeneous graph learning community to enhance its scalability and effectiveness.

3.	Comprehensive comparison experiments, visualization, and ablation studies are greatly valued. Sufficient datasets from different domains have been used to evaluate the proposed method on different downstream tasks, and the proposed method has been thoroughly discussed through experiments.

4.	I overviewed the code provided by the authors. It seems well-organized and detailed. Moreover, I tried to run the provided code with the saved checkpoints, and it can easily reproduce the results reported in the paper.

**Weaknesses:**

1. This paper aims to make the first attempt to simultaneously extract the homogeneity and the heterogeneity without meta-path in the heterogeneous graph. So, what’s the difference and relationship between the homogeneity extraction and the heterogeneity extraction in the heterogeneous graph?

2. As this paper proposes a self-supervised framework for SHGL methods, some recent related works in self-supervised learning can be added in the related work section.

3. The paper needs more proofreading and some typos should be fixed. For example,
In the second paragraph of page 4: neighbor set->neighbors set.

**Questions:**

see above

---

> ### Author Response · Authors · 2023-11-18
> **Response to Reviewer EpeM**
>
> We thank the reviewer for the careful reading and constructive discussions.
>
> > **W1**. This paper aims to make the first attempt to simultaneously extract the homogeneity and the heterogeneity without meta-path in the heterogeneous graph. So, what’s the difference and relationship between the homogeneity extraction and the heterogeneity extraction in the heterogeneous graph?
>
> Both the homogeneity extraction and the heterogeneity extraction focus on connections and information aggregation among nodes. Given a heterogeneous graph with several node types, the homogeneity mining focuses on the connectivity and information aggregation among nodes within the same class, while the heterogeneity mining focuses on the connectivity and information aggregation among nodes from different types. Moreover, in the revision, we added the formal definitions of homogeneity mining and heterogeneity mining in Appendix C.1.
>
> > **W2**. As this paper proposes a self-supervised framework for SHGL methods, some recent related works in self-supervised learning can be added in the Related Work.
>
> Thanks for your suggestions. In the revision, we added some recent works in self-supervised learning in the Related Work.
>
> > **W3**. The paper needs more proofreading and some typos should be fixed. For example, In the second paragraph of page 4: "neighbor set" should be "neighbors set".
>
> In the revision, we thoroughly proofread and fixed typos according to your suggestions.

---

### Official Review · Reviewer_h7xr · 2023-11-01

**Soundness:** 3 good
**Presentation:** 3 good
**Contribution:** 4 excellent
**Rating:** 6
**Confidence:** 3

**Summary:**

A different insight is given in this paper, compared to previous self-supervised heterogeneous graph representation learning, i.e. capture the homogeneity and heterogeneity without meta-paths. The authors have performed theoretical analysis of the proposed framework from the perspective of grouping effect and downstream tasks.  In addition, sufficient experimental results show that the proposed method achieves best results in several datasets.

**Strengths:**

(1)	The paper is clearly written and describes a novel insight and framework for the SHGL, which is rare in similar works.

(2)	Some theoretical guarantees are given to illustrate the proposed method.

(3)	Some ablation studies are conducted to verify the effectiveness of each component.

**Weaknesses:**

(1)	The proposed method uses the same projection head to map homogeneous and heterogeneous representations into the same potential space. Do the projection head have to be the same? What are the implications of using two projection heads to map homogeneous and heterogeneous representations separately?

(2)	The proposed method employs the closed-form solution of the self-expressive matrix to capture the homogeneity in the heterogeneous graph. It is desirable to clarify the process of deriving the closed-form solution so that the reader can understand the method Section more easily.

**Questions:**

The proposed method is also evaluated on the homogeneous graph datasets, e.g. Amazon-photo. On homogeneous graph datasets, does the method proposed in this paper not require data augmentation? What are the advantages of the method proposed in this paper compared to popular self-supervised methods on homogeneous graphs that require data augmentation?

---

> ### Author Response · Authors · 2023-11-18
> **Response to Reviewer h7xr**
>
> We appreciate the reviewer for the careful reading of our paper and detailed discussions.
>
> > **W1**. The proposed method uses the same projection head to map homogeneous and heterogeneous representations into the same potential space. What are the implications of using two projection heads to map homogeneous and heterogeneous representations separately?
>
> In the revision, to verify the effectiveness of the projection head, we investigated the performance of variant methods with the same projection head and different projection heads, respectively, and reported the results in Table 10 in Appendix E.6. Obviously, the variant method with the same projection head obtains the best performance. The reason can be attributed to the fact that the same projection head can map homogeneous and heterogeneous representations into the same space, so that they are comparable in the same latent space.
>
> > **W2**. The proposed method employs the closed-form solution of the self-expressive matrix to capture the homogeneity in the heterogeneous graph. It is desirable to clarify the process of deriving the closed-form solution so that the reader can understand the method Section more easily.
>
> In the manuscript, the process of deriving the closed-form solution is provided in Appendix C.2.
>
> > **Q1**. The proposed method is also evaluated on the homogeneous graph datasets, e.g., Amazon-photo. On homogeneous graph datasets, does the method proposed in this paper not require data augmentation? What are the advantages of the method proposed in this paper compared to popular self-supervised methods on homogeneous graphs that require data augmentation?
>
> Yes, the proposed method does not require data augmentation on homogeneous graph datasets. There are several advantages of the proposed method compared to popular self-supervised methods on homogeneous graphs that require data augmentation. First, the performance of existing augmentation-based methods is highly dependent on the choice of augmentation scheme, while our method does not need carefully designed augmentation techniques. Second, the self-expressive matrix used in this work is able to capture the homophily among nodes more comprehensively than other methods that use augmentation techniques.

---

### Official Review · Reviewer_6ShR · 2023-11-06

**Soundness:** 2 fair
**Presentation:** 3 good
**Contribution:** 2 fair
**Rating:** 6
**Confidence:** 4

**Summary:**

This paper proposes HERO, a self-supervised heterogeneous graph learning method that captures both homogeneity and heterogeneity in the input heterogeneous graph. The authors theoretically analyze the grouping effect provided by capturing homogeneity and the task-related information provided by capturing both homogeneity and heterogeneity. In experiments, HERO outperforms state-of-the-art baselines on both heterogeneous graph datasets and homogeneous graph datasets.

**Strengths:**

The idea of collectively and explicitly capturing homogeneity and heterogeneity for better SHGL is novel and interesting. The paper is generally well-written without obvious typos or grammatical errors.

**Weaknesses:**

1. The authors confuse homogeneity and homophily, which impairs the technical soundness of this paper. The definition of homogeneity in the paper is more like homophily. Homogeneity (single node/edge type) and homophily (connected nodes tend to have the same class label) are two different concepts.
2. Overall, many of the arguments proposed by the authors sound far-fetched. It also seems that the authors take many things for granted.
    1. Some claims of the authors are not true. For example:
        - "previous SHGL methods generally overlook or cannot effectively utilize the heterogeneity." Many existing SHGL methods, such as metapath2vec, can capture the heterogeneity. All those GNN-based SHGL methods can also capture both aspects as long as they employ heterogeneous GNN models.
        - "metapaths are employed to capture the homogeneity in the heterogeneous graph." Actually, many SHGL methods also leverage metapaths to capture the heterogeneity, such as metapath2vec, which applies DeepWalk to the node sequences (containing multiple types of nodes) generated by the pre-defined metapath.
    2. Unclear benefit of mining heterogeneity. The example given in the introduction section is hard to understand.
    4. Missing formal definitions for some important terminologies, including homogeneity, heterogeneity, and homogeneity rate.
    5. Missing derivations or theorem proof for some HERO components, including Equation (10) and Equation (11).
    6. The observations stated in Section 2.3 come with no supporting evidence.
3. Missing important baseline: metapath2vec.

**Questions:**

Please check the weaknesses above for the main issues. Here are some additional questions/comments.
1. What is the evaluation protocol for HERO and other baselines? Is it to train a linear classifier (e.g., SVM) on top of (frozen) representations learnt by self-supervised methods?
2. Metapaths can also associate two nodes with different node types. The definition of metapaths given on page 3 is therefore misleading. It's just many GNN-based methods tend to use the same-type version to construct homogeneous graphs.
3. The idea of this paper is kind of similar to DHGCN [1], which also explicitly and separately aggregates heterogeneous neighbors (heterogeneity) and metapath-guided same-type neighbors (homogeneity).
4. Definition 2.1 is currently hard to understand. The sentence "if the conditions ... hold" may need to be changed to "if $|c_{ik}-c_{jk}|\rightarrow 0$ ($\forall 1 \leq k \leq F^\prime$) hold for every $v_i, v_j$ satisfying $v_i \rightarrow v_j$ (i.e., $||x_i-x_j||_2 \rightarrow 0$)"
5. It would be better to also visualize the self-expressive matrix, to show that it indeed captures homogeneity/homophily.

[1] Saurav Manchanda, Da Zheng, George Karypis: Schema-Aware Deep Graph Convolutional Networks for Heterogeneous Graphs. IEEE BigData 2021: 480-489

---

> ### Author Response · Authors · 2023-11-18
> **Response to Reviewer 6ShR (part1)**
>
> We thank the reviewer for the careful reading of our paper and constructive comments in detail.
>
> > **W1**. The authors confuse homogeneity and homophily, which impairs the technical soundness of this paper. The definition of homogeneity in the paper is more like homophily. Homogeneity (single node/edge type) and homophily (connected nodes tend to have the same class label) are two different concepts.
>
> Thanks for the constructive comments. In the revision, we replaced "homogeneity" with "homophily".
>
> > **W2**. Overall, many of the arguments proposed by the authors sound far-fetched. It also seems that the authors take many things for granted.
>
> > **W2-1**. Some claims of the authors are not true.
> **(1)** “previous SHGL methods generally overlook or cannot eff ectively utilize the heterogeneity." Many existing SHGL methods, such as metapath2vec, can capture the heterogeneity. All those GNN-based SHGL methods can also capture both aspects as long as they employ heterogeneous GNN models. **(2)** “metapaths are employed to capture the homogeneity in the heterogeneous graph." Actually, many SHGL methods also leverage metapaths to capture the heterogeneity.
>
> **(1)** In the revision, we changed it to "most previous SHGL methods overlook or cannot effectively utilize the heterogeneity" for a more precise claim.
>
> Actually, according to previous literature [1,2,3,4], random walk based methods (e.g., metapath2vec) are generally summarized as traditional unsupervised heterogeneous graph learning methods. Compared to the currently popular self-supervised heterogeneous graph learning methods, the traditional unsupervised heterogeneous graph learning method metapath2vec ignores the rich information in the node features and thus obtains inferior performance.
>
> Moreover, as the reviewer mentioned,  the SHGL methods can capture both the homophily and heterogeneity in the heterogeneous graph as long as they employ heterogeneous GNN models. However, current mainstream SHGL frameworks (e.g., [1,2,3,4,5]) usually only use GCN to obtain node representations based on metapaths, thus **ignoring** heterogeneity. Although very few SHGL methods (e.g., HeCo [6]) employ heterogeneous GNN models, they **cannot effectively** utilize the heterogeneity. The reason is that HeCo directly aligns the heterogeneous representations and homophily representations to lose specific task-related information within each of them. In the manuscript, we discussed and summarized their differences in the Related Work.
>
> Finally, we would like to reiterate our innovative nature and contributions, which are not only simply capturing the homophily and heterogeneity like previous SHGL methods.
> First, the proposed method comprehensively captures the homophily from both the subspace and nearby neighbors without pre-defined meta-paths. However, previous methods (including metapath2vec, HeCo, and other SHGL methods) need pre-defined meta-paths, thus introducing extra expert knowledge and computational costs.
> Second, the proposed method extracts the consistent information and the specific information between homophilous and heterogeneous representations to introduce more task-related information. In contrast, most previous SHGL methods ignore the heterogeneity or cannot effectively utilize it.
>
> [1] Zhu Y, Xu Y, Cui H, et al. Structure-enhanced heterogeneous graph contrastive learning. In SDM 2022.
>
> [2] Li B, Jing B, Tong H. Graph communal contrastive learning. In WWW 2022.
>
> [3] Jing B, Feng S, Xiang Y, et al. X-GOAL: multiplex heterogeneous graph prototypical contrastive learning. In CIKM 2022.
>
> [4] Wang Z, Li Q, Yu D, et al. Heterogeneous graph contrastive multi-view learning. In SDM 2023.
>
> [5] Mo Y, Lei Y, Shen J, et al. Disentangled multiplex graph representation learning. In ICML 2023.
>
> [6] Wang X, Liu N, Han H, et al. Self-supervised heterogeneous graph neural network with co-contrastive learning. In SIGKDD 2021.
>
> **(2)** Based on the above analysis, in the revision, we changed it to ``metapaths can be employed to capture the homophily in the heterogeneous graph."
>
> > **W2-2**. Unclear benefit of mining heterogeneity. The example given in the introduction section is hard to understand.
>
> To better understand the benefits of mining heterogeneity, we give examples in  the Introduction. Here, we explain the example in detail.
>
> Considering an academic heterogeneous graph with several node types (e.g., author, paper, and subject), if two authors have the same name and similar backgrounds, e.g., two authors with the same name in the same institution.  In this case, it is difficult to distinguish these two authors. In contrast, if we utilize the heterogeneity, i.e., considering both the author nodes and the other type of nodes (e.g., each author's published papers), we can easily distinguish them with the same name from their published papers.

---

> > ### Comment · Reviewer_6ShR · 2023-11-18
> >
> > > Considering an academic heterogeneous graph with several node types (e.g., author, paper, and subject), if two authors have the same name and similar backgrounds, e.g., two authors with the same name in the same institution. In this case, it is difficult to distinguish these two authors. In contrast, if we utilize the heterogeneity, i.e., considering both the author nodes and the other type of nodes (e.g., each author's published papers), we can easily distinguish them with the same name from their published papers.
> >
> > I still find this example not convincing enough. Essentially, the problem given in your example is resolved by considering the neighborhood information (because the two authors have different neighborhoods). The "heterogeneity" is not the determining factor here.

---

> ### Author Response · Authors · 2023-11-18
> **Response to Reviewer 6ShR (part2)**
>
> > **W2-3**. Missing formal definitions for some important terminologies, including homogeneity, heterogeneity, and homogeneity rate.
>
> In the revision, we added the definitions of homophily ratio, homophily mining, and heterogeneity mining in Appendix C.1.
>
>
> > **W2-4**. Missing derivations or theorem proof for some HERO components, including Equation (10) and Equation (11).
>
> In the revision, we  added the theoretical analysis for the consistency loss in Eq. (10) and the specificity loss in Eq. (11) and reported them in Appendix C.5.
>
> Based on these theoretical analysis, we demonstrate that the consistency loss and the specificity loss extract the consistent information between homophilous representations and heterogeneous representations and maintain their specific information, respectively.
>
>
>
> > **W2-5**.  The observations stated in Section 2.3 come with no supporting evidence.
>
> To support two observations in Section 2.3, we provide more insights about them.
>
> Homophilous representations and heterogeneous representations share the same original node features, so they are intuitive to share the same information, the consistent information for short in this work.
>
> The homophilous representations focus on aggregating the information from nodes of the same class, while the heterogeneous representations focus on aggregating the information from nodes of different types. Obviously, homophilous representations and heterogeneous representations have different focuses, so we call them to contain specific information for each of them.
>
> Take the academic heterogeneous graph with several node types (e.g., paper, author, and subject) as an example, and each paper collects its abstract as the original node features. After that, the homophilous representations of each paper aggregate the information from other papers that may come from the same class, while the heterogeneous representations aggregate the information of nodes from other types (i.e., author and subject). Therefore, the homophilous representations of each paper consist of two parts, i.e., the original features (i.e., abstract) information of the paper and the information from other papers. The heterogeneous representations of each paper also consist of two parts, i.e., the same abstract information of the paper and the information from authors and subjects. In this example, the original features (i.e., abstract) information of each paper can be regarded as the consistent information, while the information from other papers and the information from authors and subjects can be regarded as the specific information within homophilous representations and heterogeneous representations, respectively.
>
> > **W3**. Missing important baseline: metapath2vec.
>
> In the revision, we added the metapath2vec as one of the baselines, and reported the results in the comparison experiments.
>
> > **Q1**. What is the evaluation protocol for HERO and other baselines? Is it to train a linear classifier (e.g., SVM) on top of (frozen) representations learnt by self-supervised methods?
>
> Yes. We follow the standard evaluation protocol according to previous self-supervised heterogeneous graph learning works [7,8,9].
>
> Specifically, we first train models with unlabeled data in a self-supervised manner and output the learned node representations. After that, the resulting representations are used to train a simple logistic regression classifier with a fixed iteration number.
>
> [7] Park C, Kim D, Han J, et al. Unsupervised attributed multiplex network embedding. In AAAI 2020.
>
> [8] Wang X, Liu N, Han H, et al. Self-supervised heterogeneous graph neural network with co-contrastive learning. In SIGKDD 2021.
>
> [9] Jing B, Feng S, Xiang Y, et al. X-GOAL: multiplex heterogeneous graph prototypical contrastive learning. In CIKM 2022.
>
> > **Q2**. Metapaths can also associate two nodes with different node types. The definition of metapaths given on page 3 is therefore misleading. It's just many GNN-based methods tend to use the same-type version to construct homogeneous graphs.
>
> In the revision, we revised the  definition of metapaths given on Page 3. Specifically, given the heterogeneous graph, the meta-path can be defined in the form of $v_{1} \stackrel{r_{1}}{\rightarrow} v_{2} \stackrel{r_{2}}{\rightarrow} \cdots \stackrel{r_{s}}{\rightarrow} v_{s+1}$.  It is a sequence of a composite relation $r_{1} \circ r_{2} \circ \cdots \circ r_{s}$ between node $v_1$ and node $v_{s+1}$, where  $s$ indicates the length of meta-path and $\circ$ denotes the composition operator.

---

> ### Author Response · Authors · 2023-11-18
> **Response to Reviewer 6ShR (part3)**
>
> > **Q3**. The idea of this paper is kind of similar to DHGCN, which also explicitly and separately aggregates heterogeneous neighbors (heterogeneity) and metapath-guided same-type neighbors (homogeneity).
>
> DHGCN is proposed to separately aggregate heterogeneity and homophily to conduct semi-supervised learning (i.e., requiring node labels), while our method is self-supervised learning (i.e., only requiring self-supervised signals). In addition, we list other differences as follows.
>
> First, DHGCN still needs meta-paths to capture the homophily in the heterogeneous graph, which requires extra expert knowledge and computational costs. In contrast, the proposed method comprehensively captures the homophily from both the subspace and nearby neighbors without pre-defined meta-paths.
>
> Second, DHGCN simply puts the homophilous representations and heterogeneous representations together without considering consistent and specific information. In contrast, the proposed method is available to extract the consistent information and the specific information between homophilous and heterogeneous representations.
>
>
> > **Q4**. Definition 2.1 is currently hard to understand. The sentence ``if the conditions ... hold" may need to be changed.
>
> In the revision, we changed Definition 2.1 based on your suggestions.
>
>
> > **Q5**. It would be better to also visualize the self-expressive matrix, to show that it indeed captures homogeneity/homophily.
>
> In the revision, to verify that the self-expressive matrix indeed captures the homophily, we visualized the self-expressive matrix heatmaps of ACM and Yelp datasets in Figure 6 in Appendix E.6, where rows and columns are reordered by node labels.
> In the heatmaps, the darker a pixel, the larger the value of self-expressive matrix weight. Based on Figure 6, the heatmaps exhibit a similar block diagonal structure with the correlation map of homophilous representations. This indicates that the self-expressive matrix assigns large weights for nodes within the same class and small weights for nodes from different classes to describe each node. Therefore, the self-expressive matrix indeed captures the homophily in the heterogeneous graph.

---

> ### Comment · Reviewer_6ShR · 2023-11-20
> **Comments after author's response**
>
> The authors have successfully addressed most of my concerns (especially the homogeneity/homophily concern). It is advised to put more effort into explaining the benefits of capturing heterogeneity. Given thorough considerations, I decide to change my rating to "6: marginally above the acceptance threshold".

---

> > ### Author Response · Authors · 2023-11-20
> > **Response to Reviewer 6ShR**
> >
> > Many thanks for your constructive advice. In our opinion, both homophily and heterogeneity are important. If homophily helps to distinguish the authors. For example, different author neighbors can distinguish two authors with the same name. In this case, heterogeneity will improve the identification confidence. Otherwise, if homophily is difficult to identify the authors, heterogeneity is the determining factor.

---

### Official Review · Reviewer_Zp3d · 2023-11-09

**Soundness:** 3 good
**Presentation:** 4 excellent
**Contribution:** 3 good
**Rating:** 8
**Confidence:** 3

**Summary:**

This paper addresses challenges in self-supervised heterogeneous graph learning. It points out two main issues: reliance on human-defined meta-paths, and underutilization of graph heterogeneity. To tackle these, the authors propose a method that captures homogeneity without predefined meta-paths using a self-expressive matrix. Additionally, they capture heterogeneity by aggregating information from different node types. Two losses are introduced to ensure consistency between homogeneity and heterogeneity and to preserve task-related information. Theoretical analysis suggests the learned representations effectively capture homogeneity and introduce more task-related information. Experimental results demonstrate the method's superiority across various downstream tasks.

**Strengths:**

S1. I really like the idea of learning homogeneity and heterogeneity without pre-defined meta-paths. It is fresh and interesting.

S2. They theoretically analyze that the learned homogeneous representations exhibit the grouping effect to capture the homogeneity, and considering both homogeneity and heterogeneity introduces more task-related information

S3. The use of a self-expressive matrix to capture homogeneity without predefined meta-paths is a creative solution, potentially reducing the need for expert knowledge. The introduction of consistency and specificity loss functions enhances the model's ability to extract and preserve task-related information.

S4.  Extensive experimental results demonstrate the effectiveness and superiority of the proposed method across various downstream tasks, validating its practical utility. The paper evaluates the proposed method thoroughly on different downstream tasks, showcasing its versatility and robust performance.

**Weaknesses:**

W1. Complexity and Computational Cost: The paper doesn't extensively discuss the computational complexity and resource requirements of the proposed method, leaving potential concerns about its scalability in large-scale applications, especially the comparison with traditional meta-path-based approaches.

W2. The whole workflow is a little complicated, which may raise concerns about reproducibility.

W3. Some important work missing, the author should discuss them in their paper, including but not limited to Self-supervised Hypergraph Representation Learning for Sociological Analysis. TKDE; Heterogeneous Hypergraph Embedding for Graph Classification. WSDM



Q1: For section 2.1 (homogeneity), I wonder what the compared results between a message-passing GNN via such a complex self-expressive matrix and the one via graph transformer (learnable self-attention)


Q2: I wonder about the scalability and efficiency of real-world datasets.


Q3: It seems the whole workflow for node classification is not an end-to-end pipeline, the author first uses their framework to pre-train the model and then they use the learned model for specific node classification. I wonder what's the detailed settings for their task head for the node classification task.


Q4. Interpretability: Although the traditional meta-path approach may have some shortcomings as mentioned by the authors,  there is no doubt that these methods are very intuitive for human understanding and explanation. I wonder how the interpretability of the learned representations is. Can the authors provide more insights into how the model's representations can be interpreted, aiding researchers and practitioners in understanding its decision-making process?

**Questions:**

please see in the above section

---

> ### Author Response · Authors · 2023-11-18
> **Response to Reviewer Zp3d (part1)**
>
> We thank the reviewer for the careful reading of our paper and constructive comments in detail.
>
> > **W1**. Complexity and Computational Cost: The paper doesn't extensively discuss the computational complexity and resource requirements of the proposed method, leaving potential concerns about its scalability in large-scale applications, especially the comparison with traditional meta-path-based approaches.
>
> In the manuscript, the time complexity of our method and the training time cost of all self-supervised heterogeneous graph learning (SHGL) methods are reported in Appendix B.2 and Appendix E.1, respectively. In the revision, we further reported the memory cost of all SHGL methods in Appendix E.1.
>
> Specifically, the time complexity of our method is linear to the sample size, while traditional meta-path-based SHGL methods (e.g., [1,2,3,4]) generally are quadratic to the sample size.
>
> In our manuscript,  we evaluated the proposed method and all SHGL methods on one large dataset of knowledge graph (i.e., Freebase with 180,098 nodes and  1,645,725 edges) in terms of classification results and the training time cost in Figure 4(a) in Appendix E.1. In the revision, we added experiments on the memory cost and report in Figure 4(b) in Appendix E.1.
> As a result,  the proposed method achieves the best performance,  minimal training time cost, and minimal memory cost on the Freebase dataset. This further verifies the scalability and efficiency of the proposed method on the large-scale dataset.
>
> [1] Park C, Kim D, Han J, et al. Unsupervised attributed multiplex network embedding. In AAAI 2020.
>
> [2] Wang X, Liu N, Han H, et al. Self-supervised heterogeneous graph neural network with co-contrastive learning. In SIGKDD 2021.
>
> [3] Jing B, Feng S, Xiang Y, et al. X-GOAL: multiplex heterogeneous graph prototypical contrastive learning. In CIKM 2022.
>
> [4] Wang Z, Li Q, Yu D, et al. Heterogeneous graph contrastive multi-view learning. In SDM 2023.
>
>
>
> > **W2**. The whole workflow is a little complicated, which may raise concerns about reproducibility.
>
> Our proposed workflow is very simple because its key part (i.e., obtaining the homophilous representations with the self-expressive matrix) was implemented with 8 lines of codes. Moreover, in the manuscript, we provided source code and datasets for reproducibility.
>
>
> > **W3**. Some important works missing.
>
> Thanks for your advice. In the revision, we discussed  these excellent works in the Related Work.
>
> > **Q1**. For section 2.1 (homogeneity), I wonder what the compared results between a message-passing GNN via such a complex self-expressive matrix and the one via graph transformer (learnable self-attention).
>
> In the manuscript, we analyzed the difference between self-expressive mechanism and self-attention mechanism in Appendix A.3. In the revision, we further compared self-attention mechanism (i.e., graph transformer)  with our self-expressive mechanism and reported the results in Table 9 in Appendix E.
>
> As a result, the self-expressive mechanism outperforms the self-attention mechanism on all datasets. The reason can be summarized as follows. First, the weights of the self-attention mechanism are non-negative, while the weights of the self-expressive mechanism are real values. Therefore, our self-expressive matrix can capture both the positive relationships among nodes within the same class and the negative relationships among nodes from different classes. Second, our self-expressive matrix is encouraged to focus on the most relevant neighbors of each sample instead of arbitrary nodes as in the self-attention mechanism. Therefore, the self-expressive matrix focuses more on the neighbors of each node within the same class and less on its faraway nodes that may come from different classes.
>
> > **Q2**. I wonder about the scalability and efficiency of real-world datasets.
>
> Please see our response in R1.
>
> > **Q3**.  It seems the whole workflow for node classification is not an end-to-end pipeline, the author first uses their framework to pre-train the model and then they use the learned model for specific node classification. I wonder what's the detailed settings for their task head for the node classification task.
>
> Yes, it is not an end-to-end pipeline. We followed the standard workflow in previous self-supervised heterogeneous graph learning works [1,2,3].
>
> Specifically, similar to [1,2,3], we first pre-train models with unlabeled data and output learned node representations, and then input the learned node representations into  a simple logistic regression for training a classifier.
>
> [1] Park C, Kim D, Han J, et al. Unsupervised attributed multiplex network embedding. In AAAI 2020.
>
> [2] Wang X, Liu N, Han H, et al. Self-supervised heterogeneous graph neural network with co-contrastive learning. In SIGKDD 2021.
>
> [3] Jing B, Feng S, Xiang Y, et al. X-GOAL: multiplex heterogeneous graph prototypical contrastive learning. In CIKM 2022.

---

> ### Author Response · Authors · 2023-11-18
> **Response to Reviewer Zp3d (part2)**
>
> > **Q4**. Interpretability: Although the traditional meta-path approach may have some shortcomings as mentioned by the authors, there is no doubt that these methods are very intuitive for human understanding and explanation. I wonder how the interpretability of the learned representations is. Can the authors provide more insights into how the model's representations can be interpreted, aiding researchers and practitioners in understanding its decision-making process?
>
> As the reviewer stated, the meta-path based methods are intuitive for human understanding and explanation. That is, two nodes connected by the meta-paths generally tend to belong to the same class and have similar representations. So does our method. For example, in our method, the nodes within the same class have similar representations. We interpreted the result of our method in Figures 3(a) and 3(b) in our manuscript.
>
> In the manuscript, we  visualized the node correlation maps of ACM and Yelp datasets in Figures 3(a) and 3(b), where rows and columns are reordered by node labels. In the correlation map, the darker a pixel, the higher the correlation between nodes. In Figures 3(a) and 3(b), the correlation maps exhibit a block diagonal structure where the nodes of each block belong to the same class. This indicates that if two nodes belong to the same class, then the correlation of their representations will be high, i.e., their representations are expected to be similar. This shows the interpretability of the learned representations.

---

### Author Response · Authors · 2023-11-18
**Summary of Author Response to All the Reviewers**

We would like to appreciate all the reviewers for their insightful comments. We revised the manuscript based on the constructive feedback and suggestions from the reviewers. We marked the contents that already existed in the manuscript (but may be missed by reviewers) in red, and those revised or newly added contents in blue in the revision.  Our key responses are summarized as follows:

**> Clarification of some arguments.**

As Reviewer 6ShR suggested, we changed the presentation of some terms to avoid confusion. For example, "homogeneity'' to "homophily'', and "homogeneous representations'' to "homophilous representations''. Moreover, we added more explanation for two observations in Section 2.3, making them more intuitive. In addition, we revised the definition of meta-path and the definition of the grouping effect according to the reviewer's suggestions.

**> Baseline method for comparison and related work for discussion.**

As Reviewer 6ShR suggested, we added metapath2vec as one of the baselines and reported the results in the comparison experiments. Moreover, as Reviewer Zp3d and Reviewer EpeM suggested, we discussed more related works about self-supervised learning and heterogeneous graph learning. In addition, we analyzed the major difference between metapath2vec and DHGCN (raised by Reviewer 6ShR) and the proposed method.

**> Detailed information of settings.**

We added the detailed information about the self-supervised workflow and evaluation protocol of the proposed method.

**> Additional theoretical analysis and definitions.**

As Reviewer 6ShR suggested, we theoretically demonstrated that the consistency loss in Eq. (10) and the specificity loss in Eq. (11) extract the consistent information between homophilous representations and heterogeneous representations and maintain their specific information, respectively. Moreover, we added the  definitions of homophily ratio, homophily mining, and heterogeneity mining.

**> Additional experiments.**

To fully explore the effectiveness of the proposed method, we added additional experiments according to the reviewers' suggestions: (1) the performance of the proposed method on the large-scale dataset (in Figure 4, Appendix E.1) to show its scalability and efficiency, (2) the comparison experiments between the self-expressive mechanism and the self-attention mechanism (in Table 9, Appendix E.3) to show the  effectiveness of the proposed method,  (3) the visualization of the self-expressive matrix (in Figure 4, Appendix E.9) to show that it indeed captures homophily, (4) the performance of variant methods using different encoders (in Table 11, Appendix E.7) to show that the proposed method is robust to different encoders, and (5) the performance of variant methods with different fusion mechanisms (in Table 12, Appendix E.8) to show the effectiveness of the concatenation mechanism.

**> Summary.**

We thank all the reviewers again for the detailed and constructive review. It seems that all the reviewers have agreed with the novelty of the proposed method, and that most of the concerns are raised to part of the arguments, and experiments. We hope our clarification, additional experiments, and additional theoretical analysis in this revision could address all of your concerns. Please let us know if you have any  questions or concerns.

---

### Meta-Review · Area_Chair_YBev · 2023-12-07

**Metareview:**

In this paper, the authors proposed a  self-supervised method to address some challenging issues in heterogeneous graph learning. There were some concerns about the concepts, arguments, and experimental results raised by the reviewers in their original reviews. After rebuttal, most of the issues have been addressed. Overall,  this paper presents a nice work for heterogeneous graph learning.

**Justification For Why Not Higher Score:**

Some reviewers are still not very convinced by the experiments.

**Justification For Why Not Lower Score:**

I think this paper deserves to be published.

---

### Decision · Program_Chairs · 2024-01-16

Accept (poster)